# Higher-order structure and proteoforms of co-occurring C4b-binding protein assemblies in human serum

Tereza Kadavá [ID] [1,2], Johannes F Hevler [ID] [1,2], Sofia Kalaidopoulou Nteak [ID] [1,2], Victor C Yin [ID] [1,2], Juergen Strasser [ID] [3], Johannes Preiner [ID] [3] & Albert JR Heck [ID] [1,2 ✉]

## Abstract

The complement is a conserved cascade that plays a central role in the innate immune system. To maintain a delicate equilibrium preventing excessive complement activation, complement inhibitors are essential. One of the major fluid-phase complement inhibitors is C4b-binding protein (C4BP). Human C4BP is a macromolecular glycoprotein composed of two distinct subunits, C4BPα and C4BPβ. These associate with vitamin K-dependent protein S (ProS) forming an ensemble of co-occurring higher-order structures. Here, we characterize these C4BP assemblies. We resolve and quantify isoforms of purified human serum C4BP using distinct single-particle detection techniques: charge detection mass spectrometry, and mass photometry accompanied by high-speed atomic force microscopy. Combining cross-linking mass spectrometry, glycoproteomics, and structural modeling, we report comprehensive glycoproteoform profiles and full-length structural models of the endogenous C4BP assemblies, expanding knowledge of this key complement inhibitor's structure and composition. Finally, we reveal that an increased C4BPα to C4BPβ ratio coincides with elevated C-reactive protein levels in patient plasma samples. This observation highlights C4BP isoform variation and affirms a distinct role of co-occurring C4BP assemblies upon acute phase inflammation.

**Keywords** Complement; C4b-Binding Protein; Mass Spectrometry-Based Techniques; Integrative Structural Modeling
**Subject Categories** Immunology; Proteomics; Structural Biology

## Introduction

The complement cascade is a central part of the innate immune system, capable of pathogen recognition, opsonization, and clearance (Sjöberg et al, 2009; Noris and Remuzzi, 2013). This highly-conserved proteolytic pathway is tightly regulated by a variety of activator and inhibitor proteins to maintain a delicate equilibrium and prevent complement dysregulation (Ojha et al, 2019). One of the major complement regulators is C4b-binding protein (C4BP), a dominant fluid-phase inhibitor (Fig. 1A) (Ermert and Blom, 2016). This large (~600 kDa) acute-phase serum glycoprotein is primarily known as an inhibitor of the classical and lectin pathways due to its interaction with C4b (Gigli et al, 1979). However, C4BP also interacts with C3b and mediates the decay of an alternative complement pathway (Blom et al, 2003a). Principally, C4BP acts as an essential cofactor for serine protease factor I (FI) (Fukui et al, 2002), which cleaves complement factors C3b and C4b, to render them inactive (iC3b and iC4b). These inactivated forms cannot form C3 and C5 convertases and, therefore, cannot further trigger the complement cascade (Blom et al, 2003a; Ziccardi et al, 1984).

Human serum C4BP consists of disulfide-linked C4BPα and C4BPβ chains that carry eight and three complement control protein (CCP) domains (Fig. 1B), respectively (Barnum, 1991). Additionally, the coagulation inhibitor vitamin K-dependent protein S (ProS) interacts non-covalently with C4BPβ (Dahlbäck and Stenflo, 1981). These three protein chains assemble into higher-order structures (HOS), referred to as C4BP isoforms or variants (Sánchez-Corral et al, 1995; García et al, 1995). The reported isoforms possess either 7 C4BPα, 1 C4BPβ, and ProS (α7β1+ProS); 6 C4BPα, 1 C4BPβ, and ProS (α6β1+ProS); or 7 C4BPα (α7) chains (Fig. 1C) (Sánchez-Corral et al, 1995; Dahlbäck et al, 1983). The α7 variant is sometimes referred to as C4BP(β−) and, accordingly, ProS-bound isoforms containing C4BPβ are noted as C4BP(β+). Despite the compositional difference, all C4BP variants can fulfill a complement-inhibiting role (Dahlbäck and Hildebrand, 1983).

A full picture of the C4BP assembly remains elusive despite several attempts to characterize its HOS and infer structure-function relationship (Ermert and Blom, 2016; Dahlbäck et al, 1983; Hofmeyer et al, 2013; Jenkins et al, 2006; Buffalo et al, 2016). Initial structural studies characterizing C4BP suggested a flexible spider-like assembly with either seven or eight "arms" (Dahlbäck et al, 1983; Perkins et al, 1986). More recently, the C-terminal region of C4BPα was identified as the core required for C4BP chain oligomerization and HOS formation (Kask et al, 2002), with the 7

[1]Biomolecular Mass Spectrometry and Proteomics, Bijvoet Center for Biomolecular Research and Utrecht Institute for Pharmaceutical Sciences, University of Utrecht, Padualaan 8, Utrecht 3584 CH, the Netherlands. [2]Netherlands Proteomics Center, Padualaan 8, Utrecht 3584 CH, the Netherlands. [3]University of Applied Sciences Upper Austria, 4020 Linz, Austria. ✉E-mail: a.j.r.heck@uu.nl

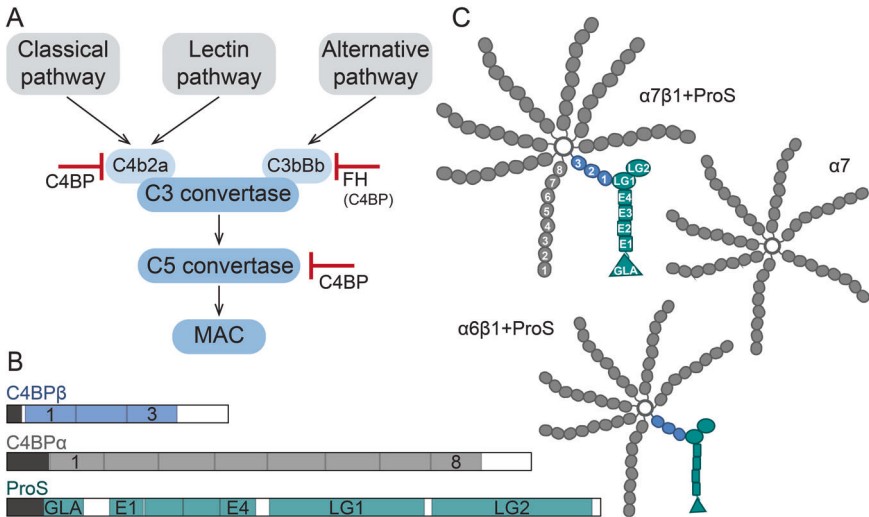

**Figure 1. C4b-binding protein (C4BP) function, subunit architecture, and its proposed higher-order structures.**

(A) Simplified scheme of the complement cascade activation with highlighted checkpoints controlled by C4BP. (B) Two subunits are forming the C4BP higher-order structures in human serum, namely C4BPα in gray and C4BPβ in blue. Complexing partner ProS is shown in teal. Propeptides and signal peptides are visualized in black. Complement control protein (CCP) domains of C4BPα and C4BPβ are numbered from the N- to C-terminus 1–8 and 1–3, respectively. ProS is composed of the N-terminal gamma-carboxy-glutamate domain (GLA), four EGF-like domains (E1–E4), and two Laminin G-like domains (LG1–2). (C) Cartoon representation of proposed co-occurring isoforms of human serum C4BP: α7β1+ProS, α6β1+ProS, and 7α.

C4BPα chains that form the core of the C4BP(β−) variant being further resolved by X-ray crystallography (Hofmeyer et al, 2013). Still, a high-resolution full-length structure of the co-occurring C4BP assemblies present in human serum remains intractable, likely due to the protein's high flexibility and compositional heterogeneity (Dahlbäck et al, 1983). In particular, key features of C4BP(β+), attachment of the β-chain to the oligomerization core, the C4BPβ–ProS interaction, and post-translational modifications of C4BP remain uncharacterized.

Here, we aim to provide a deeper understanding of the co-occurring variants of human C4BP by employing an integrative approach. We combine state-of-the-art mass spectrometry (MS)-based techniques: cross-linking MS (XL-MS), glycoproteomics, and single molecule native charge detection MS (CDMS), with high-speed atomic force microscopy (HS-AFM), and mass photometry (MP), to gain insights into the HOS and glycoproteoforms of C4BP, aiming to build full-length structural models. Furthermore, we set out to capture the co-occurring C4BP assemblies in human plasma and their variation (Sánchez-Corral et al, 1995; García et al, 1995) at the protein level. For that, we examine several serum and plasma proteomics datasets, following levels of all C4BPα, C4BPβ, and ProS under normal conditions and during acute phase inflammation. We also analyze serum size-exclusion chromatography (SEC) LC-MS to validate a tight co-elution of C4BPα, C4BPβ, and ProS in the high molecular weight (MW) fractions.

## Results and discussion

### Resolving and quantifying co-occurring C4b-binding protein assemblies

Human serum C4BP is composed of C4BPα and C4BPβ subunits assembled with ProS into several co-occurring HOS. Here, we set

out to resolve and further characterize those species, analyzing C4BP from pooled healthy donor human serum purified by Complement Technology, Inc. Before proceeding, we first examined the purity, composition, and nativity of the C4BP sample using several orthogonal approaches. Bottom-up proteomics analysis confirmed that the acquired sample is predominantly composed of C4BPα, C4BPβ, and ProS (Fig. EV1A), corresponding to the native C4BP assembly (Dahlbäck and Stenflo, 1981; Dahlbäck et al, 1983). The most abundant contaminants identified were known C4BP interacting partner C4b (C4A and C4B genes) (Dahlbäck et al, 1983; Scharfstein et al, 1978) and MBL2, but their relative abundance was below 5%. Next, we confirmed that the C4BP sample is physiologically active, capable of interacting with C4b and facilitating FI-mediated cleavage of C4b (Fig. EV1B,C).

Although the nativity and composition of the C4BP sample were verified, the findings did not provide a complete description of the C4BP variants. To capture and resolve the native C4BP isoforms, we next utilized two single-particle detection-based techniques, namely CDMS and MP. These techniques have emerged as valuable approaches for resolving large and heterogeneous biomolecular assemblies (Young et al, 2018; Wörner et al, 2020; Deslignière et al, 2023). Additionally, both aforementioned techniques can be performed under non-denaturing conditions, preserving the non-covalent C4BP(β+)–ProS interaction.

In measurements of native C4BP (Fig. 2A,B), both techniques revealed two distinct major high MW and one minor high MW distribution of particles. Based on the determined masses, the highest MW distribution can be assigned to the α7β1+ProS isoform (633 ± 22 kDa CDMS; 630 ± 29 kDa MP). We detected a second similarly abundant population ~80 kDa lower in mass (553 ± 20 kDa CDMS; 553 ± 22 kDa MP). Both techniques also exposed a third minor population of α6β1 (481 ± 20 kDa CDMS; 487 ± 22 kDa MP). Notably, all populations detected were

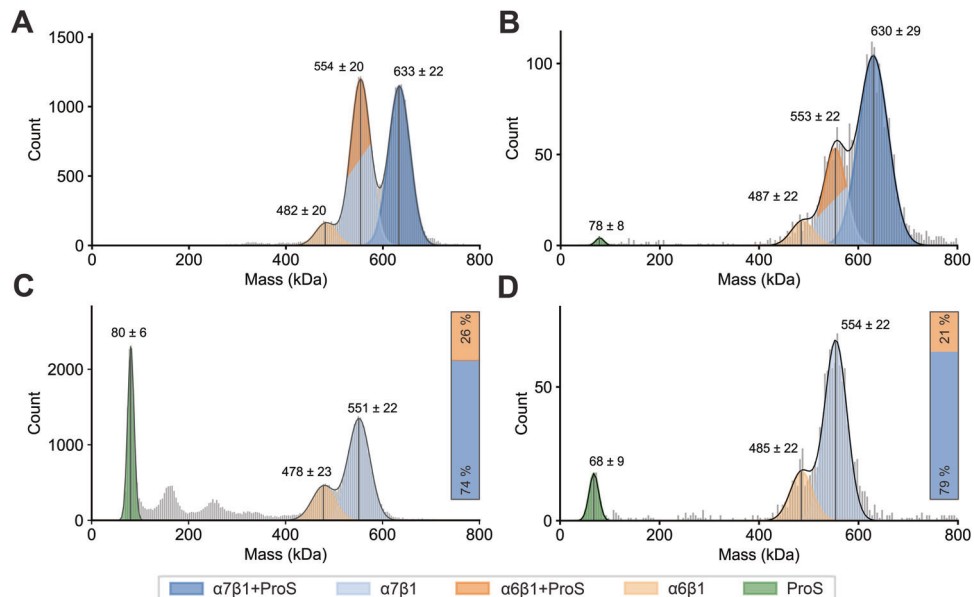

**Figure 2. Resolving and quantifying the human serum C4BP variants by charge detection mass spectrometry (CDMS) and mass photometry (MP).**

(**A**) CDMS of native C4BP, (**B**) MP of native C4BP, (**C**) CDMS of acidified C4BP, (**D**) MP of acidified C4BP. Native C4BP (**A, B**) exhibited two major populations corresponding to α7β1 + ProS (dark blue) and overlapping α6β1+ProS (light blue)/α7β1 (dark orange), as well as a small portion of unbound ProS (green) in MP. The C4BP samples, obtained by acidification (**C, D**), show ProS dissociation (green), enable isoform differentiation, and highlight dominant C4BP(β+) variants. Bars in (**C, D**) show isoform abundance of α7β1 (blue) and α6β1 (orange) in the sample, as quantified for denatured CDMS and MP, respectively. Of note, both CDMS and MP were optimized for the detection of the C4BP assembly (~400–800 kDa). Therefore, the quantification of lower MW species, such as monomeric ProS, is more ambiguous. Source data are available online for this figure.

separated by roughly 80 kDa. Yet, due to the close MW of C4BPα and ProS, we were unable to unambiguously assign the 80 kDa difference, and thus cannot distinguish α7β1 and α6β1+ProS variants in the second population. We suggest the presence of the α7β1, as we also detected the minor population corresponding to the α6β1 isoform. We attribute this observation to ProS dissociation, supported by the presence of monomeric ProS in the MP measurements (Fig. 2B).

To fully distinguish between α6β1+ProS and α7β1 and resolve the C4BP variants present in human serum, we recorded measurements under denaturing conditions. The experiments aimed to disrupt the non-covalent interaction of C4BP(β+) with ProS by acidifying the C4BP sample while preserving the disulfide bonds linking C4BPα and C4BPβ (Hillarp and Dahlbäck, 1988). This (Fig. 2C,D) revealed only two distinct high MW populations, corresponding to the C4BP(β+) variants: α7β1 (551 ± 22 kDa CDMS; 554 ± 22 kDa MP) and α6β1 (478 ± 23 kDa CDMS; 485 ± 22 kDa MP). Both histograms also showed, as expected, dissociated monomeric ProS (80 ± 6 kDa CDMS; 68 ± 9 kDa MP). Dissociation of ProS is further underlined by an ~80 kDa shift to lower mass for both of the two high MW populations, detected under native conditions (Fig. 2A,B). Moreover, a difference in the abundance of the two high MW populations was observed. This apparent discrepancy further supports our hypothesis of the overlapping α6β1+ProS and α7β1 in the native C4BP samples (Fig. 2A,B). CDMS and MP analysis revealed C4BP(β+) variants dominating the C4BP healthy serum sample, as quantified and visualized in Fig. 2C,D. By contrast, our results did not indicate a substantial population of the 7α variant in normal human serum C4BP samples.

## Exploring C4b-binding protein higher-order structures

The CDMS and MP results clearly displayed co-occurring C4BP isoforms in healthy human serum, with the predominant forms being the C4BP(β+) variants: α7β1+ProS and α6β1+ProS (Fig. 2). Next, we aimed to structurally characterize those C4BP assemblies with a multi-faceted approach combining XL-MS, glycoproteomics, HS-AFM, and structural modeling.

## Cross-linking mass spectrometry

Aiming to build a structural model of C4BP, we first set out to gain insight into the interfaces responsible for C4BP(β+) formation. To do so, we utilized XL-MS (Steigenberger et al, 2020; Iacobucci et al, 2020; Graziadei and Rappsilber, 2022), employing two complementary cross-linking chemistries, the amine-reactive NHS-ester DSS and the carboxyl-to-amine coupling DMTMM (Leitner et al, 2014). We then analyzed the cross-linked peptides with liquid chromatography (LC)-MS/MS.

The XL-MS results confirmed C4BPβ attachment to the C4BPα oligomerization core at the proteins' C-termini, as evidenced by several DSS and DMTMM cross-links (Fig. 3A,B). Among several C4BPα intra-links in the C-terminal region, we also observed self-links of C4BPα K595. This underlines the interaction of multiple C4BPα C-termini to form the oligomerization core. Interestingly, we also observed C4BPβ DMTMM interlinks between K204 and E228 residues localized in the core region. Exposing the C4BPβ–ProS interface, the XL-MS results (Fig. 3B,C) suggested an interaction mediated by the N-terminal region of the C4BPβ

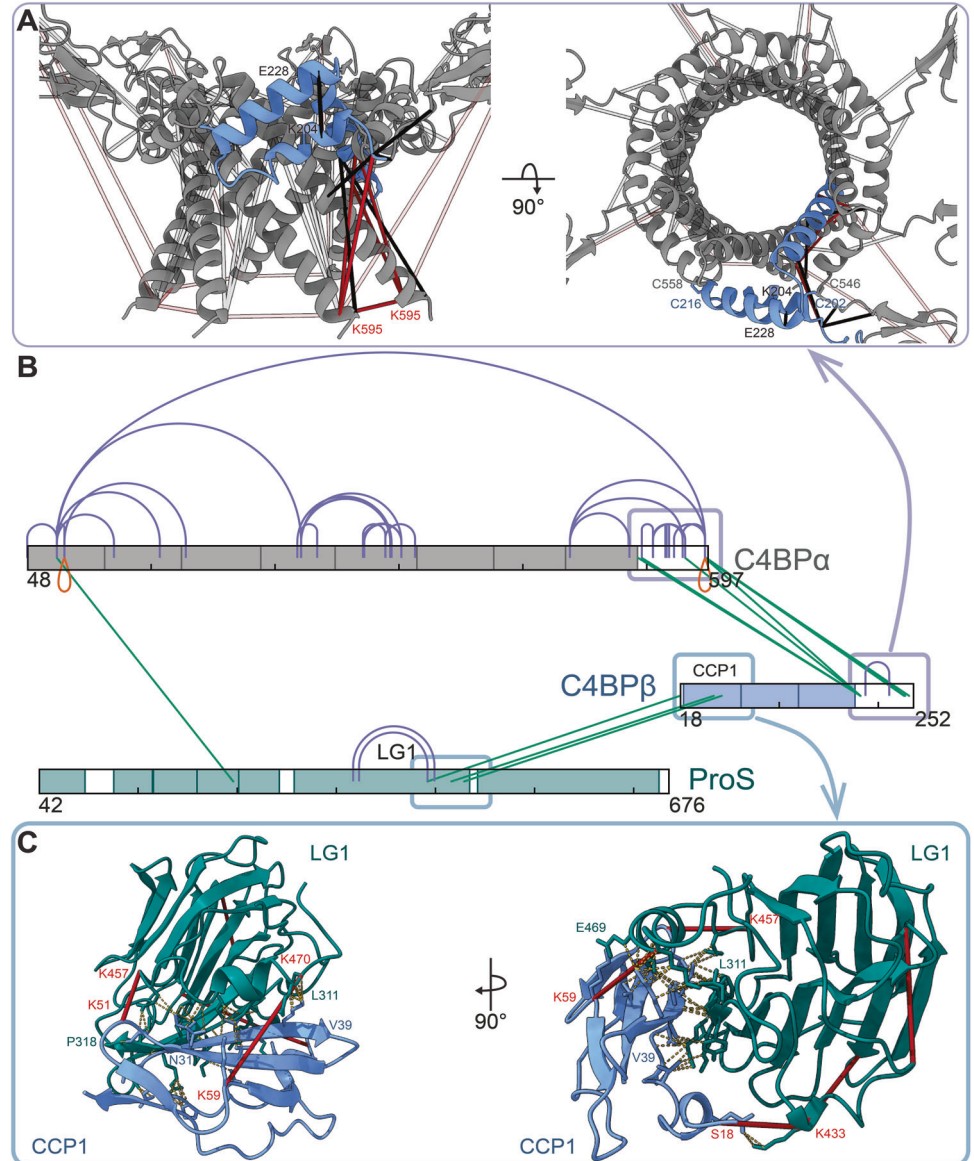

**Figure 3. Cross-linking mass spectrometry highlights interaction interfaces of the C4BP assembly.**

(A) The structural model of the α7β1 core displays seven C4BPα (gray) with C4BPβ (blue) inserted in a helix-hairpin-helix conformation with detected DMTMM (black) and DSS (red) cross-links. To highlight C4BPβ involving restraints, C4BPα intra-protein cross-links (except for the K595 self-link) are visualized as semi-transparent. Visualized were the shortest possible cross-links with an allowed 2 Å difference. (B) An overview of the observed XL-MS restraints. XL-MS revealed C4BPβ CCP1 interaction with LG1 of ProS and highlighted the assembly of C4BPα and C4BPβ at the C-terminus of both chains. Intra- (purple), inter- (green), and self- (orange) cross-links of C4BPα (gray), C4BPβ (blue), and ProS (teal) are color-annotated. Visualized were cross-links detected in two out of three experimental replicates. (C) The proposed C4BPβ–ProS interface. The structural model highlights interactions between the C4BPβ CCP3 (blue) and the ProS LG1 (teal) with the DSS cross-links (in red) and resulting interacting residues connected by yellow dashed lines. The generation of the structural models is described below in the full-length glycosylated models of the C4b-binding protein section. Source data are available online for this figure.

CCP1 and C-terminal region of the laminin G-like domain 1 (LG1) of ProS. DSS cross-links were identified between the N-terminus (S18, N-term after the propeptide cleavage) of C4BPβ linked to the ProS K433, C4BPβ K51 linked to ProS K457, C4BPβ K49, and ProS K429. Of note, the XL-MS data presented (Fig. 3B) clearly display more C4BPα cross-links compared to the ProS and C4BPβ chains, specifically C4BPα interlinks. This is likely caused by the nature of C4BP assembly, which carries either six or seven times more copies of C4BPα than of C4BPβ and ProS.

The XL-MS data provided clear structural constraints and insights into the C4BP HOS. Yet it also revealed a few so-called overlength cross-links (Appendix Fig. S1) mainly involving the N-terminal CCP1 domain of the C4BPα linked to other regions of C4BPα (CCP3, CCP4, and C-terminal oligomerization core) and ProS EGF-like domain 3 (Fig. 3B). This observation was further highlighted by C4BPα CCP1 K77 self-links (Fig. 3B, orange) and is likely reflective of the inherent flexibility of the C4BPα "arms" as elegantly earlier visualized by EM and modeled by using SAXS data (Dahlbäck et al, 1983; Perkins et al, 1986).

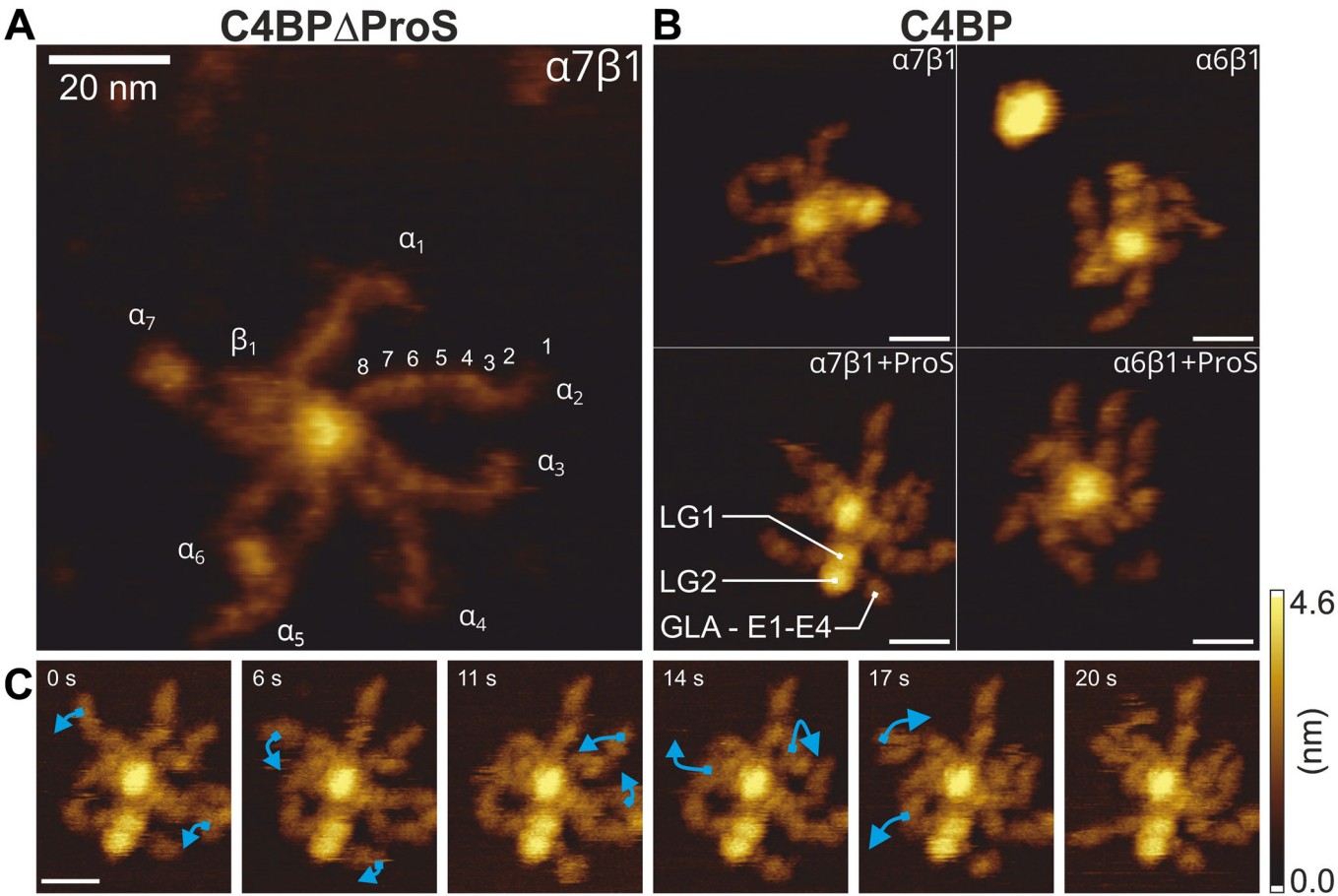

**Figure 4. C4BP higher-order structures and their flexibility visualized by high-speed atomic force microscopy in a strong immobilization buffer.**

(A) C4BP α7β1 (C4BPΔProS sample). A central core with a diameter of ~5 nm surrounded by 7 C4BPα "arms" and a shorter C4BPβ chain. The C4BPα, C4BPβ, and 8 CCP domains of α2 were annotated. (B) Representative images of the four major species identified in human serum C4BP sample (α6β1 and α7β1 variants with or without ProS). (C) HS-AFM time series following a single C4BP molecule in a strong immobilization buffer highlighting the flexibility of C4BPα. Domains moving between frames are indicated by blue arrows. All scale bars correspond to 20 nm. Source data are available online for this figure.

## High-speed atomic force microscopy

To visualize the C4BP variants and assess their flexibility and possible conformations, we utilized high-speed atomic force microscopy (HS-AFM). HS-AFM is a technique that allows direct imaging of highly dynamic biomolecules under near-physiological conditions (Ando et al, 2001; Preiner et al, 2014; Strasser et al, 2019). Here, we analyzed native human serum C4BP, and a sample depleted of ProS: C4BPΔProS (Fig. 4), analogously to the CDMS and MP experiments described above (Fig. 2). The spider-like structure of C4BP was readily apparent when imaging C4BPΔProS in a strong immobilization buffer (Fig. 4A). We observed a large circular central core surrounded by six or seven long "arms" (C4BPα subunits) consisting of eight CCP domains and a single shorter one (C4BPβ subunit). Native human serum C4BP displayed the same structure, but a subpopulation representing 71% (compared to 74 and 68% for MP and CDMS, respectively) of all detected particles from the observed complexes presented with two globular domains localized close to the oligomerization core (Fig. 4B). Those were, based on earlier reported observations, assigned as LG1 and LG2 domains of non-covalently attached ProS

(Dahlbäck et al, 1983). Additional structures corresponding to the EGF-like and GLA domains of ProS were also resolved. The remaining 29% of complexes in the native C4BP sample lacked ProS, resulting in four C4BP species α6β1, α6β1+ProS, α7β1, and α7β1+ProS as initially resolved by their distinctive masses by CDMS and MP (Fig. 2). Of note, no ProS-decorated C4BP variants were observed in the ProS-depleted sample, further confirming ProS depletion from the sample.

C4BP displayed a high degree of flexibility even in a strong immobilization buffer, which facilitates strong electrostatic attachment between protein and mica. This flexibility is exemplified by the range of conformations apparent in Fig. 4A,B. Furthermore, continuous observation of an individual C4BP HOS revealed time-resolved structural dynamics of the C4BPα (Fig. 4C; Movie EV1). This flexibility was observed predominantly for the N-terminal CCP domains of C4BPα, as well as for the ProS EG and GLA domains. The described structural dynamics may account for the above-described detected overlength cross-links observed, particularly for N-terminal CCPs of the α-chain (Fig. 4; Appendix Fig. S1). Further, we highlighted the flexibility of native human serum C4BP in a weak immobilization buffer (Movie EV2) and demonstrated

the non-covalent nature of the C4BP–ProS interaction (Movie EV3). The weaker electrostatic attachment in these experiments allowed the C4BP "arms" to move freely, leading to the C4BP structures rapidly changing positions and conformations. Unfortunately, these rapid movements resulted in less-resolved structures.

## Full-length glycosylated models of C4b-binding protein

Attempting to complete the picture of C4BP HOS, we used the XL-MS data along with the HS-AFM results as a base to generate α7β1+ProS (Fig. 6A) and α6β1+ProS (Fig. 6B) structural models. First, the C4BPα and ProS protein chains were truncated based on XL-MS identified interfaces. Specifically, C-termini of C4BPα forming the oligomerization core with the full-length C4BPβ and LG domains of ProS interacting with CCP1 of C4BPβ (Fig. 3) were used as an input for AlphaFold-Multimer (Evans et al, 2022). The resulting models displayed the insertion of C4BPβ into the oligomerization core in a helix-hairpin-helix conformation (Figs. EV2). Such C4BPβ orientation is supported by the XL-MS data, specifically by the interlink connecting E228 and K204 of C4BPβ (Fig. 3A,B). Furthermore, the proposed orientation is consistent with disulfide bridges connecting C4BPα and C4BPβ, as both cysteine pairs exhibited distances shorter than 2.05 Å for the α7β1+ProS variant (Appendix Fig. S2B,C). The proposed oligomerization core model is fully compatible with the HS-AFM results showing 4.5 nm core thickness (Appendix Fig. S2A).

Finally, to complete the models of the C4BP glycoprotein assemblies, we combined the core model with predicted structures of full-length C4BPα and ProS chains, resulting in models of C4BP. However, our models lacked glycans, and C4BP is a glycoprotein with all its protein chains harboring several N-glycosylation motifs. As protein glycosylation plays an eminent role in protein structure, interactions, and function (Cumming, 1991; Watanabe et al, 2020), we aimed to get insight also into the C4BP glycoproteoforms. To do so, we used peptide-centric glycoproteomics, exposing, identifying, and quantifying post-translational modifications occurring on C4BPα, C4BPβ, and ProS (Fig. 5). The data revealed three C4BPα (N221, N506, N528), five C4BPβ (N64, N71, N98, N117, and N154), and three ProS (N499, N509, and N530) N-glycosylation sites. Even though a variety of glycan modifications was observed for each site, $HexNAc_4Hex_5Neu5Ac$ and $HexNAc_4Hex_5Neu5Ac_2$ were dominant. Both correspond to complex biantennary glycans with either one or two sialic acids attached. The only exception was ProS N509, which was decorated mostly by the triantennary sialylated complex N-glycans $HexNAc_5Hex_6Neu5Ac_2$ or $HexNAc_5Hex_6Neu5Ac_3$. Consistent with our results, biantennary glycans decorating liver-synthesized serum proteins, as well as a smaller extent of C4BPα sialylation, were previously reported (Ritchie et al, 2002), although not described in a site-resolved manner as we have here.

We then used the site-specific information and visualized the most abundant N-glycosylation for each site on the structural models (Fig. 6). The resulting full-length glycosylated models of dominant C4BP(β+) variants correspond to the "spider-like" HOS as visualized by HS-AFM (Fig. 4) and previously by negative stain EM (Dahlbäck et al, 1983). Further, they are consistent with XL-MS-identified interfaces (Fig. 3).

The final, complete models provide unique insights into the C4BP assembly. They display 6 or 7 C4BPα chains (for α6β1+ProS or α7β1+ProS, respectively), with C4BPβ assembled into the core structure by C-terminal alpha-helixes and CCP domains of both protein chains spreading out from the core. Independently, HS-AFM (Fig. 4) and XL-MS (Fig. 3) indicated high flexibility of the C4BPα N-terminal CCP domains, especially of the CCP1. In contrast, focusing on the CCP7 and 8 located closer to the oligomerization core, we found fewer overlength cross-links and more even subunit spacing in HS-AFM. Inspecting the full-length model, C4BPα N506 and N528 glycosylation sites are found close to the oligomerization core. The respective glycans face towards the adjacent α chains and display high occupation rates (Fig. 5). Thus, we propose that the CCP8 glycosylation might play a role in maintaining even C4BPα spacing close to the oligomerization core region, helping to preserve its observed rigidity. More distant from the C4BPα and C4BPβ oligomerization core we observed highly flexible "arms", formed by N-terminal CCP domains in C4BPα. These C4BP regions are crucial for complement inhibition, facilitating the FI-mediated decay of complement factors C4b (Blom et al, 2001) and C3b (Blom et al, 2003a). Therefore, it seems that C4BP flexibility is tightly dictated by its function. The dynamic C4BPα chains are thought to fulfill several functions, including, in particular, bringing several proteins (e.g., FI and C4b/C3b on a cell surface (Ermert and Blom, 2016)) in close proximity and also triggering structural rearrangement of C4/C3 essential for their cleavage by FI (Blom et al, 2003b). Interestingly, the CCP3 domain situated in this key C4BPα region harbors N221 glycosylation. Yet the possible role of this modification, which occupies the vast majority of C4PBα chains, has not been considered in this context.

Human C4BP HOS includes C4BPβ in contrast to some other species (Blom et al, 2004). Compared to the C4BPα chain, C4BPβ carries just three CCP domains, with the N-terminal CCP1 of the C4BPβ interacting with the LG1 domain of ProS. As shown in Figs. 3C and EV2, we suggest that the C4BPβ-ProS interaction is predominantly mediated by an antiparallel beta-sheet formed between C4BPβ N31-V39 and ProS L311-P318. The proposed interface is supported by previous mutation experiments highlighting the importance of hydrophobic residues in the V16–F45 region of the C4BPβ for the ProS binding (Webb et al, 2001). Further inspecting the C4BPβ–ProS interface in the structural models of C4BP (Fig. 6), two N-glycans are attached to the CCP1 of the C4BPβ and the LG2 domain of ProS carries three N-glycosylation modifications. Importantly, previous results suggested that neither the three ProS glycosylation sites (Lu et al, 1997) nor the C4BPβ glycosylation (Blom et al, 2004) plays a role in binding. The C4BP models are compatible with those findings, showing that the C4BPβ-ProS interface is spatially well separated from the N-glycosylation sites on either protein chains (Fig. 5).

Our structural models, however, contrast with a previous attempt to visualize the C4BPβ–ProS interaction (Blom et al, 2004), where C4BPβ CCP1 was proposed to bind between the LG1 and LG2 domains of ProS. Nevertheless, we predict such binding to be highly unlikely. Firstly, CCP1 binding between the LG domains would not be compatible with our XL-MS results. Secondly, the full-length structural model shows that such binding would likely be highly dependent on the N-glycans covering the respective regions of both ProS and C4BPβ, in opposition to previous

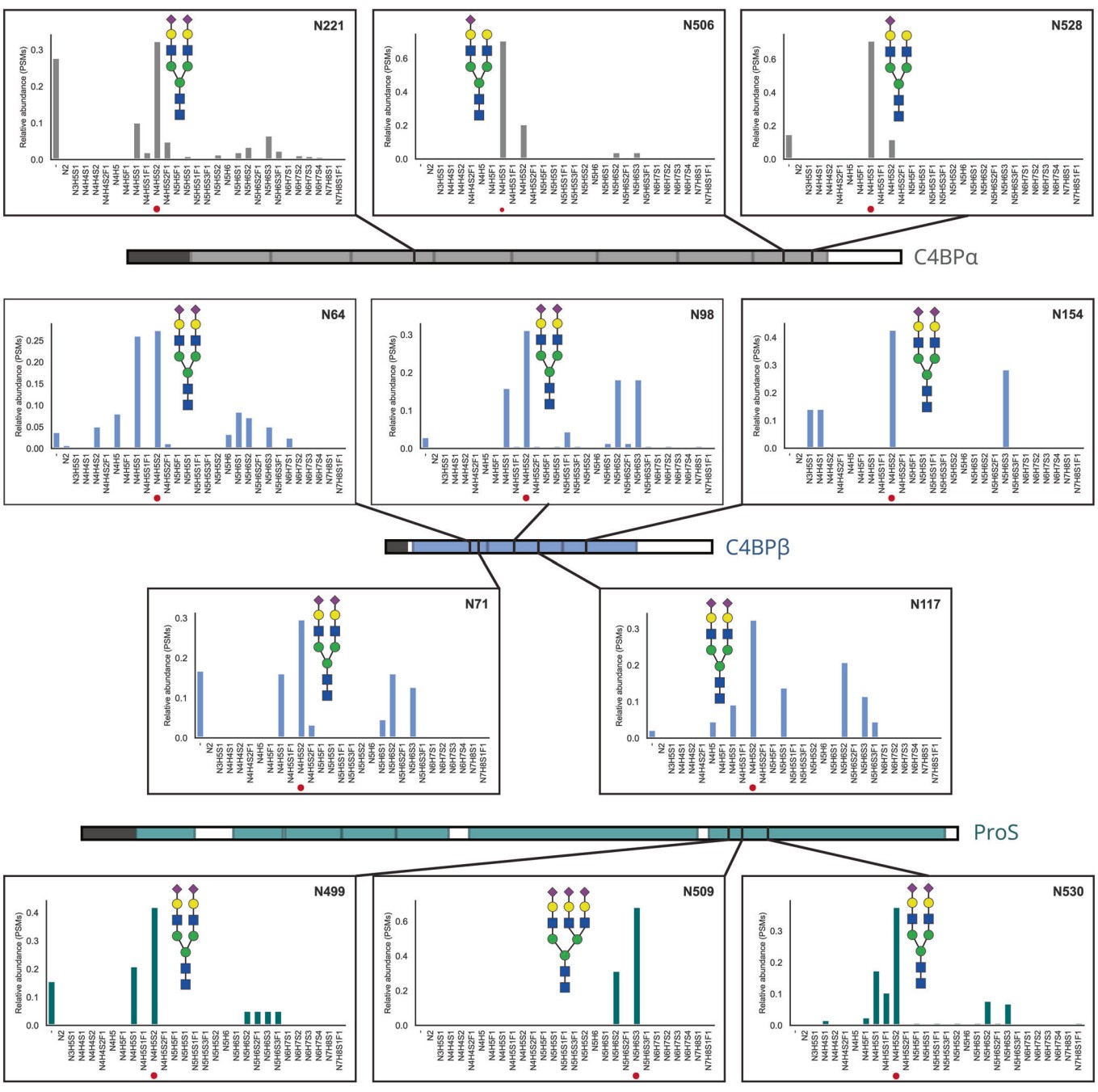

**Figure 5. N-glycosylation of C4BP.**

The figure shows C4BPα (gray), C4BPβ (blue), and ProS (teal) N-glycosylation as identified by peptide-centric glycoproteomics and quantified based on PSMs. Modifications were visualized as bar charts for each site modified with schematic depictions of the most abundant N-glycan moiety. The data revealed the occupation of the majority of C4BP sites by HexNAc4Hex5 with either one or two sialic acids corresponding to the complex biantennary glycans. The vast majority of sites displayed a low abundance of unoccupied sites (−), except for C4BPα (N221 and N528) and C4BPβ (N71), which exhibited more than 20% unmodified residues. The following abbreviations were used: N–HexNAc, H–Hex, F–dHex, and S–NeuAc. Source data are available online for this figure.

observations (Blom et al, 2004; Lu et al, 1997). Hence, we suggest that the C4BP structural models presented in Fig. 6 may provide an improved description of the C4BPβ–ProS interaction, as they are compatible both with previous observations (Blom et al, 2004; Webb et al, 2001; Lu et al, 1997) and the current XL-MS data (Fig. 3).

## Analyzing native C4b-binding protein confirms isoform variation

The CDMS and MP data, accompanied by the full-length structural models, clearly highlighted the co-occurrence of distinct C4BP stoichiometries. Notably, α6β1+ProS and α7β1+ProS variants sharing

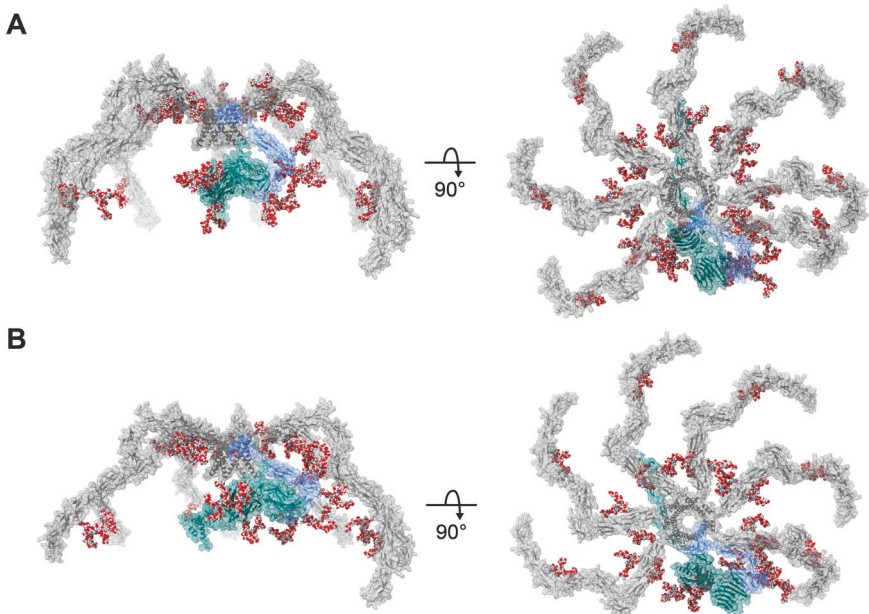

**Figure 6.  Full-length, glycosylated, spider-like structural models of C4b-binding protein assemblies.**

The models display the C4BP(β+) variants co-occurring in human serum: (**A**) α7β1+ProS and (**B**) α6β1+ProS, with C4BPα (gray), C4BPβ (blue), and ProS (cyan). The models are decorated by the most abundant glycan moieties, which were experimentally detected for each N-glycosylation site (Fig. 5). N-glycans are visualized as spheres representing atoms (H–white, C–gray, N–blue, and O–red). Source data are available online for this figure.

similar structures (Fig. 6), were found to be dominant. To expand our analyses beyond the purified C4BP sample, we assessed human C4BP directly from serum or plasma (Fig. EV3). We examined C4BPα and C4BPβ abundance in a deposited proteomics dataset of more than 650 human plasma samples (Demichev et al, 2021). In comparison with fibrinogen α-chain (FGA) and fibrinogen β-chain (FGB) (Fig. EV3A), which are known to form a stable complex in a 1:1 ratio (Kollman et al, 2009), this analysis revealed steady ratios of C4BPα to C4BPβ relative intensities, affirming their organization in HOS (Fig. EV2B). Yet, the results suggested more variability in C4BP HOS compared to the fibrinogen complexes. To further validate the assembly of the C4BP chains with ProS, we examined a SEC LC-MS dataset of human serum samples (Doorduijn et al, 2022). This analysis (Fig. EV2C) demonstrated a tight co-elution of C4BPα, C4BPβ, and ProS in high MW fractions, followed by monomeric ProS eluting later in low MW fractions.

Highlighting the presence of different C4BP HOS in human serum, the distinct functions of those variants have not been fully described. Therefore, we set out to characterize native human C4BP isoforms and their variation along with ProS abundance directly from plasma samples at the protein level. We examined label-free quantification (LFQ) data-independent acquisition (DIA)-MS plasma proteomics results of four distinct donors (Fig. 7; Appendix Fig. S3) (Kalaido-poulou Nteak et al, 2024). Two of these were healthy controls (C1 and C2), and the other two were patients who experienced several bacterial infections after a kidney transplant (P1 and P2).

Specifically, we used the longitudinal plasma proteomics data to explore levels of C4BP subunits (C4BPα, C4BPβ), ProS, and C-reactive protein (CRP). The latter is a hallmark protein widely used to monitor an acute phase reaction, which is typically triggered by a bacterial infection (Gewurz et al, 1982). While the CRP levels of the two healthy donors remained relatively low and stable over the time points

analyzed (Fig. 7), both kidney transplant patients showed highly elevated levels of CRP for certain time points, corresponding to the time of clinically diagnosed infection (Appendix Fig. S3).

To gain insight into C4BP variant abundance during acute phase inflammation, we quantified and monitored ratios of C4BPα to C4BPβ (Fig. 7). This ratio should fall between 6 and 7, reflecting the co-occurrence of C4BP(β+) variants detected by MP, CDMS, and HS-AFM (Figs. 2, 4). As anticipated, the healthy donor samples (C1 and C2) exhibited stable ratios in this range, affirming the C4BP(β+) co-occurrence in healthy individuals. Moreover, the observed ratios were stable, yet somewhat unique for both healthy donors, agreeing with previous observations (Sánchez-Corral et al, 1995; García et al, 1995). In contrast, we observed an elevated C4BPα/C4BPβ ratio during an acute phase. The CRP levels show that patient P1 underwent an acute phase inflammation around T1 and T6–T8, while patient P2 experienced inflammation peaking at T2. For all those time points, the C4BPα/C4BPβ far exceeded 7, clearly indicating C4BP isoform variation. Moreover, the highly elevated C4BPα/C4BPβ hints at the presence of a 7α variant without the β chain. On the other hand, we observed unexpectedly low C4BPα/C4BPβ levels at T0 for both patients. This timepoint corresponds to patients suffering from severe kidney failure, preceding a kidney transplant (Appendix Fig. S3), a state characterized by proteinuria and accompanied by dramatic changes in serum proteome (Dubin et al, 2023). This might explain the low C4BPα/C4BPβ ratios in those samples, which are further supported by a recent proposal that C4BP is an early biomarker of kidney damage (Rhode et al, 2023).

While clearly capturing the C4BP isoform variation over the acute phase, we further focused on the ProS levels. We found steady levels of unbound ProS in all samples analyzed, for both healthy controls and acute phase patients, as shown in Fig. 7. As the C4BP-bound ProS is

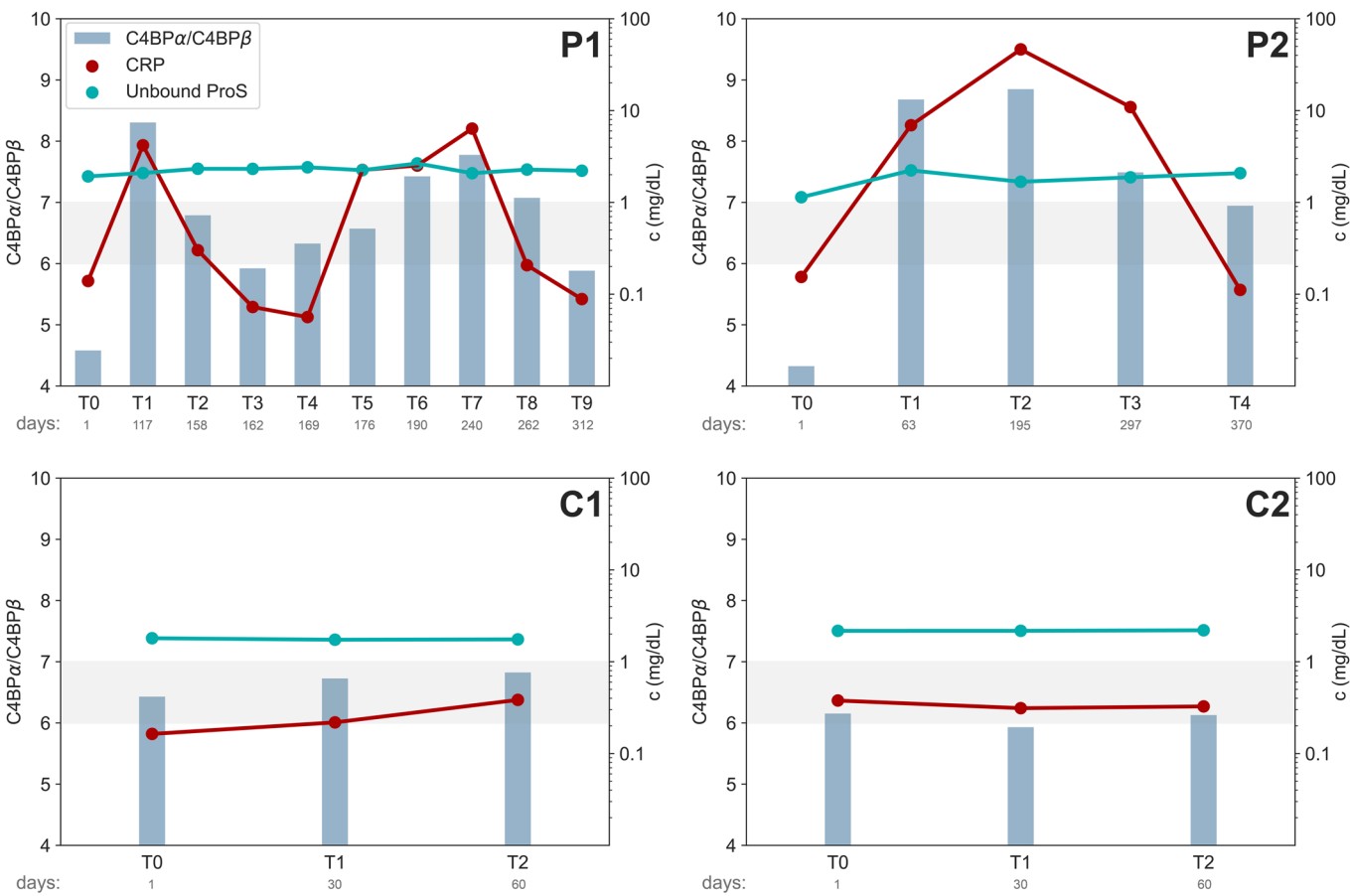

**Figure 7. The overall composition of C4BP changes during acute phase events.**

The ratio of C4BPα to C4BPβ (blue bars) is shown on the primary y-axis (left). The concentrations of unbound ProS (teal) and C-reactive protein (red) are shown on the secondary y-axis (right). The data covers two healthy donors (C1 and C2) and two kidney transplant patients (P1 and P2). The gray area highlights the expected C4BPα/C4BPβ range of values (6 to 7) corresponding to solely α6β1(+ProS) and α7β1(+ProS) variants, as resolved by single-particle detection techniques. Source data are available online for this figure.

incapable of coagulation inhibition (Ermert and Blom, 2016), this observation brings new experimental evidence for possible C4BP variation explanation. Our results support the role of C4BP isoform variation in preserving stable levels of unbound ProS, in agreement with previous hypotheses (Ermert and Blom, 2016; Blom et al, 2004). Furthermore, we suggest that healthy plasma C4BP is dominated by the C4BP(β+) variants interacting with ProS. During acute phase inflammation, C4BP 7α abundance increases, fulfilling the essential complement inhibition role (Dahlbäck and Hildebrand, 1983), yet lacking the C4BPβ and therefore not interacting with ProS. This leads to stable levels of unbound ProS, further explaining this unique complement-coagulation crosstalk.

## Methods

### Materials and chemicals

If not otherwise stated, all experiments were conducted using Milli-Q H₂O generated by the IQ 7003 system (Merck). GluC, trypsin, and chymotrypsin were acquired from Promega. LysC was acquired from

Wako. Ammonium bicarbonate, chloroacetamide (CAA), dithiothreitol (DTT), iodoacetamide (IAA), sodium deoxycholate (SDC), Tris, Tris(2-carboxyethyl)phosphine (TCEP), and urea were acquired from Merck. SDS-PAGE was performed using Criterion XT Bis-Tris Precast gels, respective electrophoretic cells, XT Sample Buffer, and Precision Plus Protein Dual Color Standard, all purchased from Biorad. Ammonium acetate, formic acid (FA), PBS, and trifluoroacetic acid (TFA) were acquired from Thermo Fisher Scientific Inc. LC-MS solvents A (0.1% v/v FA in H₂O), and B (80% acetonitrile (ACN) v/v, 0.1% FA v/v) were acquired from Biosolve Chimie.

Data presented in this publication were partially post-processed by in-house developed R (R Core Team, 2021), or Python scripts relying on the following libraries and packages: NumPy (Harris et al, 2020), pandas (McKinney, 2010), Matplotlib (Droettboom et al, 2015), seaborn (Waskom, 2021), Pyteomics (Goloborodko et al, 2013), and SciPy (Virtanen et al, 2020).

### C4BP purification

C4BP sample used for CDMS, MP, XL-MS, and glycoproteomics analyses was purified from pooled plasma of more than 16 healthy

donors (Complement Technology, Inc). The procedure included several non-denaturing chromatographic steps without the involvement of denaturing agents and Ba-citrate precipitation. As shown in Fig. EV1, the sample contained predominantly C4BP HOS (C4BPα, C4BPβ, and ProS). Contaminants forming less than 5% majorly consisted of known C4BP interacting partners most likely retained due to the non-denaturing purification procedure.

## Orbitrap-based charge detection mass spectrometry

The C4BP sample (Complement Technology, Inc) was buffer exchanged into 1 M aqueous ammonium acetate, pH = 7.4 using an Amicon Ultra-0.5 mL centrifugal unit with 50 kDa molecular weight cut-off (MWCO) (Sigma-Aldrich). The buffer exchange was performed in five centrifugation cycles (9000 × g, 5 min), adding 400 µL of 1 M ammonium acetate after each centrifugation. The sample was subjected to a final centrifugation cycle (9000 × g, 10 min) and further used for the CDMS analysis at ~1 µM concentration. A denatured sample with dissociated ProS was prepared by FA addition (pH ~ 2). FA was used mainly for its compatibility with MS analysis. The same mass distribution of urea and FA-treated C4BP sample was confirmed by MP (Appendix Fig. S4).

The CDMS spectra were recorded in positive ionization mode on an Orbitrap Q Exactive UHMR mass spectrometer (Thermo Fisher Scientific Inc), as previously described elsewhere (Wörner et al, 2020; Ebberink et al, 2022). Briefly, the sample was ionized using an in-house prepared gold-coated borosilicate emitter. First, the instrument settings were adjusted for optimal ion transmission. CDMS measurement was performed with noise level 0, injection time 1 ms, and trapping gas pressure 1. The CDMS spectra were acquired with 512 ms (100,000 resolution at 400 $m/z$) and 1024 ms (200,000 resolution at 400 $m/z$) transients for native and denatured (acidified) C4BP samples, respectively. The collected data were converted to mzXML by RawConverter (He et al, 2015) and processed by in-house developed Python scripts (Wörner et al, 2020). CDMS calibration was performed using known protein standards, resulting in a factor of 14.713 used to convert normalized single ion intensities to charges of respective ions. The relative abundance of C4BP isoforms was quantified from the multiple Gaussian fit, comparing single ion events corresponding to respective populations.

## Mass photometry

The MP results for native and denatured C4BP (Complement Technology, Inc) were obtained on the OneMP instrument (Refeyn Ltd.) in a medium field of view. Sample carrier slides were prepared in-house from microscope coverslips (24 mm × 50 mm; Paul Marienfeld GmbH) and reusable cell culture gaskets (Grace Biolabs). The data were acquired using AcquireMP (Refeyn Ltd.). Before the measurement, 12 µL of PBS (pH = 7.4) was pipetted on the sample carrier and used for the automatic focusing procedure. Immediately after the focusing, 3 µL of the sample was added, resulting in 16 nM C4PB (Complement Technology, Inc) either in PBS (pH = 7.4) for the native sample or PBS (pH = 2 adjusted with FA, to achieve similar conditions as for the CDMS experiment) for the denatured C4BP sample. Data were collected for 120 s with a scan rate of 100 frames per second.

The instrument was calibrated using a standard protein mixture (73, 149, 483, and 800 kDa) diluted in PBS. Raw movies were processed and calibrated in DiscoverMP (Refeyn Ltd.). Calibrated events were exported and further analyzed using an in-house Python script (den Boer et al, 2022). The relative C4BP isoform composition was quantified from the multiple Gaussian fit, similarly as described for CDMS results.

## C4BP activity assay

C4BP activity in mediating C4b degradation by FI was tested, as described previously (Blom et al, 2003b). Briefly, C4BP 0.1 mg/mL was mixed with C4b 0.1 mg/mL, and FI 0.01 mg/mL for SDS-PAGE or 0.04 mg/mL for MP (all proteins were purchased from Complement Technology, Inc) in a buffer: 50 mM Tris-HCl, pH = 7.4, 150 mM NaCl. Two additional samples without either C4BP or FI were prepared as a control. All samples were incubated for 1.5 h at 37 °C and subsequently analyzed by reducing SDS-PAGE and MP.

Prior to the SDS-PAGE analysis were the samples heated (5 min, 95 °C) in the XT sample buffer with 25 mM DTT. The separation was performed according to vendor-provided protocol. The resulting SDS-PAGE gel was stained by Imperial Protein Stain (Thermo Fisher Scientific Inc).

MP analysis of C4BP-C4b complexes was performed similarly as described above. Samux MP instrument (Refeyn Ltd.) was calibrated using a standard protein mixture (335, 670a, and 1340 kDa). The samples were measured in 50 mM Tris-HCl, pH = 7.4, and 150 mM NaCl at 10 µM concentration. The data were acquired for 60 s with a scan rate of 100 frames per second.

## Cross-linking mass spectrometry

The C4BP sample (Complement Technology, Inc, 0.5 g/l, PBS; pH = 7.4) was cross-linked using 1 mM disuccinimidyl suberate (DSS, Thermo Fisher Scientific Inc, final 5% DMSO in the sample) or 5 mM 4-(4,6-Dimethoxy-1,3,5-triazin-2-yl)-4-methylmorpholinium chloride (DMTMM, Thermo Fisher Scientific Inc) for 30 min, at the room temperature (RT). Immediately after, the reactions were quenched by 1 M Tris, pH = 8.5, to achieve 25 mM final concentration and incubated (30 min, RT). The resulting cross-linked samples were analyzed by MP (Appendix Fig. S5B–D). Each of the triplicate samples was subsequently separated by a reducing or non-reducing SDS-PAGE (Appendix Fig. S5A), together with a control incubated without cross-linker in 5% DMSO. Selected bands were excised and processed using an in-gel digestion protocol (Shevchenko et al, 2006). Briefly, the excised bands were cut in cubes, washed with MQ, reduced by 6.5 mM DTT, 50 mM ammonium bicarbonate, pH = 8.5 (1 h, 60 °C), and alkylated 55 mM IAA, 50 mM ammonium bicarbonate, pH = 8.5 (30 min, RT, in dark). The alkylation was followed by washing the pieces two times with 50 mM ammonium bicarbonate, pH = 8.5. Importantly, each step was followed by shrinking the pieces by 100% ACN (15 min, RT). The last dehydration step was followed by submerging the gel pieces in trypsin solution (3 ng/µL) and incubating (90 min, on ice). The last dehydration step was followed by submerging the gel pieces in trypsin solution (3 ng/µL) and incubating (90 min, on ice). Further, the excess enzyme solution was removed, substituted with 50 mM ammonium bicarbonate,

pH = 8.5, and incubated (16 h, 37 °C). Next, the supernatant with digested peptides was collected, and gel pieces shrank with 100% ACN. Both supernatants were combined, and the solvent was fully evaporated. Dry peptides were dissolved in 2% FA before the LC-MS/MS analysis.

The samples were analyzed on an UltiMate 3000 UHPLC system (Thermo Fisher Scientific Inc) coupled to an Orbitrap Exploris 480 mass spectrometer (Thermo Fisher Scientific Inc). The cross-linked peptides were analyzed in a 90 min gradient, first trapped on Acclaim Pepmap 100 C18 (5 mm × 0.3 mm, 5 μm, Thermo Fisher Scientific Inc) for 1 min with a flow rate of 30 μL/min solvent A (0.1% FA v/v in $H_2O$), and separated on an in-house packed analytical column (Reprosil 2.4 μm, 75 μm × 50 cm) with a flow rate of 0.3 μL/min using gradient starting from 9% solvent B (80% acetonitrile (ACN) v/v, 0.1% FA v/v) at 0–1 min, 13% B at 2 min; 44% B 67 min, 55% B 72 min, 99% B 75 min, and equilibrated for next run in 9% B at 80–90 min. The Orbitrap Exploris 480 collected MS1 scan every second with 60,000 resolution, 375–1600 m/z range, standard AGC target, and automatic injection time. Subsequent MS2 scans were performed in data-dependent acquisition mode with a 1.4 m/z isolation window with 14 s dynamic exclusion after one measurement. The charge states 2–6+ were selected for fragmentation with 28% NCE, and mass spectra were collected at a 15,000 resolution from 120 m/z.

### Cross-linking mass spectrometry data analysis

The obtained raw files were analyzed by Proteome Discoverer 3.0 (Thermo Fisher Scientific Inc) with incorporated XlinkX node for cross-linked peptides search (Liu et al, 2015; Klykov et al, 2018). The XL-MS search was performed using a subset FASTA. The C4BP protein sample used for peptide-centric glycoproteomics digested by trypsin (SI) was searched against reference Homo Sapiens Uniprot proteome (downloaded 2022/09/20 from Uniprot (The UniProt Consortium, 2021)) MaxQuant 2.1.4 (Tyanova et al, 2016) with 1% FDR, carbamidomethylation (C) set as fixed modification, oxidation (M) and acetylation (protein N-term) as variable modifications, trypsin/P with two missed cleavages was set as protease. The resulting non-contaminant protein groups with IBAQ $>9.5 \times 10^6$ were used as a subset database for the XL-MS search. Further, the MaxQuant search results were used to assess commercial C4BP sample purity. The results were filtered for non-contaminant protein groups, and their iBAQ values were normalized to 100%. The protein groups with relative abundance >1% were visualized in a pie chart. Both C4 isotypes C4B and C4A were grouped and named as C4b (Feucht et al, 1986).

The non-cleavable XlinkX search was performed for both DSS (K-K) and DMTMM (E|D–K), also allowing N-terminus linkage. Static modification was defined as carbamidomethyl (C), dynamic modifications were set as oxidation (M), and acetylation (protein N-term), and trypsin with a maximum of three missed cleavages was selected as protease. The FTMS and Precursor Mass tolerance was set to 50 ppm and the FDR threshold to 1 and 5% for DSS for DMTMM, respectively. The XL-MS search outputs for distinct gel bands were grouped for each gel line. Cross-links were considered only if they had an XlinkX score >40 and were identified in at least two of three experimental replicates. The filtered data for C4BP were visualized by xiView (Martin Graham et al, 2019) and on the structural models of C4BP using XMAS (Lagerwaard et al, 2022)

for ChimeraX (Pettersen et al, 2021). Visualized were the shortest possible cross-links with a 2 Å length difference allowed.

### High-speed atomic force microscopy

The C4BPΔProS sample was prepared similarly as previously described (de Cordoba et al, 1983). The C4BP sample (Complement Technology, Inc) was diluted in 7.2 M urea in PBS, pH = 7.4 (final concentration), and loaded on an Amicon Ultra-0.5 centrifugal unit with MWCO 100 kDa (Sigma-Aldrich). Next, three centrifugal cycles were performed (5 min, 8000 × g, 20 °C), adding 0.4 mL of 8 M urea in PBS, pH = 7.4 in PBS after each cycle. Finally, eight centrifugal cycles were performed (10 min, 8000 × g, 4 °C), adding 0.4 mL of PBS, pH = 7.4 after each cycle. The resulting sample was analyzed using MP (Appendix Fig. S4).

HS-AFM (RIBM, Japan) was conducted in tapping mode at RT in buffer, with free amplitudes of 1.5–2.5 nm and amplitude set points larger than 90%. Silicon nitride cantilevers with electron-beam deposited tips (USC-F1.2-k0.15, Nanoworld AG), nominal spring constants of 0.15 N/m, resonance frequencies around 500 kHz, and a quality factor of ~2 in liquids were used. All samples were incubated on freshly cleaved muscovite mica at 6 μg/mL in weak immobilization buffer (75 mM NaCl, 10 mM Tris, 10 mM $MgCl_2$, pH = 7.4) for 6 min, followed by three washing steps using the same buffer. Imaging was performed either in the same buffer or in a strong immobilization buffer (75 mM NaCl, 10 mM Tris, 10 mM $NiCl_2$, pH = 7.4), as indicated. Images were processed using Gwyddion 2.62 and ImageJ (NIH).

### Peptide-centric glycoproteomics

To gain coverage of all C4BP assembly components, three different proteolytic digestion workflows were employed, using either trypsin + LysC, trypsin + GluC, or chymotrypsin. First, the C4BP samples (Complement Technology, Inc) were reduced and alkylated (1% SDC (w/v), 10 mM TCEP, 40 mM chloroacetamide, 50 mM Tris; pH 8.5) for 30 min, 20 °C. Hereafter, the samples were diluted tenfold, resulting in a final 50 mM ammonium bicarbonate, pH = 8.5. The proteolytic digestion was performed either using GluC (1:50 (w/w), 37 °C, 4 h) followed by the addition of trypsin (1:50 (w/w), 37 °C, 16 h), chymotrypsin (1:50 (w/w), 25 °C, 16 h), or trypsin and LysC (1:50 (w/w)/1:100 (w/w), respectively, 37 °C, 16 h). The digestion was stopped by acidification with TFA (0.5% (v/v) final concentration), and the SDC precipitate was separated by centrifugation (16,000 × g, 15 min). The supernatant was desalted on Oasis HLB μElution Plate (Waters Corp) using a vendor-provided protocol. The solvent from desalted peptides was fully evaporated, and for the LC-MS/MS analysis were the dry peptides dissolved in 2% FA (v/v). The peptides were analyzed using an UltiMate 3000 UHPLC system (Thermo Fisher Scientific Inc) coupled to an Orbitrap Exploris 480 or Fusion mass spectrometer (Thermo Fisher Scientific Inc). The peptides were analyzed in a 60 min gradient, first trapped on Acclaim Pepmap 100 C18 (5 mm × 0.3 mm, 5 μm, Thermo Fisher Scientific Inc) for 1 min with a flow rate of 30 μL/min solvent A (0.1% FA v/v in $H_2O$), and separated on an in-house packed analytical column (Reprosil 2.4 μm, 75 μm × 50 cm) with a flow rate of 0.3 μL/min using gradient starting from 9% solvent B (80% acetonitrile (ACN) v/v, 0.1% FA v/v) at 0–1 min, 13% B at 2 min; 44% B 42 min, 99% B

45 min, and equilibrated for a next run in 9% B at 50–60 min. MS1 scans were acquired with 120,000 resolution, 300–2000 m/z range, 400,000 AGC target, and maximum injection time of 50 ms on Orbitrap Fusion or in 350–2000 m/z range and standard AGC Target on an Orbitrap Exploris 480. For each sample the eluting peptides were analyzed by three different MS/MS methods either using higher-energy collisional dissociation (HCD) using NCE 29%, oxonium ion-triggered stepped HCD (stHCD) using NCE 10, 25, and 40%, or oxonium ion-triggered electron-transfer/higher-energy collisional dissociation (EThcD) as previously described elsewhere (Gazi et al, 2023). Briefly, MS2 scans prioritizing higher charge states and lower m/z were performed in data-dependent acquisition mode with a 3 m/z isolation window and 30 s dynamic exclusion after one measurement. Charge states 2–8+ were selected for fragmentation and mass spectra were collected at a 60,000 resolution in a 120–4000 m/z.

The resulting files were searched by Byonic (Protein metrics) using non-specific digestion, precursor mass tolerance 10 ppm, fragment mass tolerance 20 ppm, defining HCD fragmentation type for HCD and stHCD, and EThcD for EThcD. Carbamidomethyl (C) was set as a fixed modification and phospho (S, T, Y), oxidation (M, W), pyro-glu (E), and pyro-gln (N), acetyl (K, N-term), were set as rare1 variable modifications. N-glycans (SI) were set as a common2 variable modification and the FASTA file used for the search included all C4BP protein chains (C4BPα, C4BPβ, and ProS). The search outputs were combined and filtered for scores>150 and |Log prob|>1.5 and further manually assessed. The outputs were grouped by the modification site, quantified based on PSMs, and visualized by site-specific bar charts.

## Structural model building

The structural models of C4BP(β+) variants were assembled from a core consisting of 6 or 7 (for α6β1 + ProS and α7β1 + ProS variants, respectively) C-terminal C4BPα sequences, full-length C4BPβ and two C-terminal LG domains of ProS generated by AlphaFold-Multimer (version 2.2.0) (Evans et al, 2022). ProS and C4BPα chains were truncated accordingly to interfaces identified in the XL-MS experiment. Furthermore, the truncations were in agreement with the previously proposed interaction interfaces (Hofmeyer et al, 2013; Webb et al, 2001). Both AF runs resulted in 25 models, from which the first 5 were assessed using an in-house developed Python script generating a predicted alignment error (PAE) plot and a predicted local-distance difference test (pLDDT) plot. For both α6β1 + ProS and α7β1 + ProS, the best-ranked (rank_0) model was selected and further used to build a full-length model. To do so, full-length C4BPα and ProS were modeled by the AlphaFold2 (Jumper et al, 2021). Further, signal peptides of all three protein chains were removed, and the truncated protein chains were attached to the core models. Clashes were refined using Modeller model loops (Webb and Sali, 2016) for ChimeraX (Pettersen et al, 2021). The most prevalent N-glycosylation, detected by glycoproteomics, was attached to the full-length models using the CHARMM-GUI glycan reader and modeler (Park et al, 2019). The glycan structures that were attached, represent the most common N-glycan linkage isoforms for serum glycoproteins (Stanley et al, 2022) and were not confirmed experimentally.

## Bottom-up proteomics of inflamed patients

First, 1 µL of plasma sample from each individual at each timepoint was diluted by 24 µL buffer (1% SDC, 10 mM TCEP, 10 mM Tris pH = 8.5, 40 mM chloroacetamide, with cOmplete™ mini EDTA-free protease inhibitor cocktail (Roche)) and boiled (95 °C, 5 min). For the proteolytic digestion were samples diluted in 50 mM ammonium bicarbonate, pH = 8, and the digestion was performed using trypsin and LysC in 1:50 (w/w) and 1:75 (w/w), respectively. The digestion was quenched by FA (10% (v/v) final concentration), and the resulting peptides were desalted using the AssayMap Bravo platform (Agilent Technologies) with the corresponding AssayMap C18 reverse-phase column. The eluate was fully evaporated and resolubilized in 1% FA.

Approximately 1 µg of peptides for each sample was analyzed on an Orbitrap Exploris 480 mass spectrometer (Thermo Fisher Scientific) operated in DIA mode coupled to an Ultimate 3000 liquid chromatography system (Thermo Fisher Scientific Inc). First, the peptides were trapped on an Acclaim Pepmap 100 C18 (5 mm × 0.3 mm, 5 µm, Thermo Fisher Scientific Inc) trap and separated on a 50 cm reversed-phase column packed in-house (Poroshell EC-C18, 2.7 µm, 50 cm × 75 µm; Agilent Technologies). Proteome samples were eluted over a linear gradient of a dual-buffer setup with buffer A (0.1% (v/v) FA) and buffer B (80%(v/v) ACN, 0.1%(v/v) FA) ranging from 9 to 44% B over 65 min, 44–99% B for 3 min, and maintained at 9% B for the final 5 min with a flow rate of 300 nL/min. DIA runs consisted of an MS1 scan at 60,000 resolution at m/z 200 followed by 30 sequential quadrupole isolation windows of 20 m/z for HCD MS/MS with detection of fragment ions in the orbitrap (OT) at 30,000 resolution at m/z 200. The m/z range covered was 400–1000 and the Automatic Gain Control was set to 100% for MS and 1000% for MS/MS. The injection time was set to "custom" for MS and "auto" for MS/MS scans.

The raw files were analyzed using DIA-NN Software (version 1.8) (Demichev et al, 2020) in library-free mode. The digestion enzyme was set as Trypsin with one missed cleavage tolerated. Carbamidomethylation (C) and oxidation (M) were allowed. The precursor false discovery rate threshold was set to 1%. Protein grouping was done by protein and gene names, and cross-run normalization was RT-dependent. All other settings were kept at the default values. The gene report matrix from DIA-NN was used for downstream analysis, and quantification was based on the MaxLFQ values of the unique and razor peptides with 1% filtering at both peptide and protein group levels. Next, the median log MaxLFQ values per protein group from the mass spectrometry experiments were converted into plasma protein concentrations. For this conversion, as described previously (Völlmy et al, 2021), a linear regression of the log-transformed MaxLFQ with 22 known reported average concentrations of plasma proteins (A2M, B2M, C1R, C2, C6, C9, CFP, CP, F10, F12, F2, F7, F8, F9, HP, KLKB1, MB, MBL2, SERPINA1, TFRC, TTR, VWF) was performed (Schenk et al, 2008). The median concentrations from the injection replicates were used. All data manipulation was carried out in the R programming language. The C4BPα to C4BPβ ratios were calculated as the ratio of respective protein concentrations. The unbound ProS levels were calculated by subtracting C4BPβ from total ProS abundance, assuming a high affinity of C4BPβ and ProS (Dahlbäck et al, 1990).

## Serum/plasma C4BP analysis

Two datasets were examined to access C4BP directly from human serum and plasma. The first dataset included more than 650 human plasma samples analyzed by DIA-MS proteomics (Demichev et al, 2021). The dataset was used to access FGB, FGA, C4BPα, and C4BPβ abundances. Linear regression was used to examine the protein chain ratios.

The second dataset included healthy human pooled donor serum SEC LC-MS (Doorduijn et al, 2022). It was used to examine C4BPα, C4BPβ, and ProS abundances and their elution profiles in serum SEC. Triplicate data were summed and plotted with standard deviation as a rolling mean over three fractions.

## Data availability

All LC-MS/MS data (XL-MS and glycoproteomics) have been deposited to the ProteomeXchange Consortium via the PRIDE (Perez-Riverol et al, 2021) partner repository with the dataset identifier PXD047679. CDMS and MP data were available on figshare (https://figshare.com/s/89129b360493d4d42723) as well as the full-length structural models of C4BP (https://figshare.com/s/9d2144c0bbe2c06d9232).

The source data of this paper are collected in the following database record: biostudies:S-SCDT-10_1038-S44318-024-00128-y.

## Peer review information

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

## Acknowledgements

We acknowledge funding through the Dutch Research Council (NWO) for the Netherlands Proteomics Centre through the X-omics Road Map program (project 184.034.019). A.J.R.H. acknowledges further support from the Netherlands Organization for Scientific Research (NWO) through the Spinoza Award SPI.2017.028. J.P. acknowledges support by the Federal State of Upper Austria as a part of the FH Upper Austria Center of Excellence for Technological Innovation in Medicine (TIMed Center) and the Austrian Science Fund (FWF, Grant No. P33958 and P34164). We acknowledge Andris Jankevics for help with XL-MS data processing, Karli R. Reiding for suggestions regarding the glycoproteomics, Evolène Deslignière for the CDMS calibration, and Franziska Völlmy for generating the longitudinal proteomics results. We acknowledge Dr. Nenoo Rawal (Complement Technology, Inc) for providing details regarding C4BP sample purification. Additionally, we would like to thank Amber D. Rolland, Marta Šiborová, and Jan Fiala for their suggestions and advice.

## Author contributions

**Tereza Kadavá**: Conceptualization; Formal analysis; Investigation; Writing—original draft; Writing—review and editing. **Johannes F Hevler**: Conceptualization; Investigation; Methodology; Writing—review and editing. **Sofia Kalaidopoulou Nteak**: Formal analysis; Investigation; Methodology; Writing—review and editing. **Victor C Yin**: Investigation; Methodology; Writing—review and editing. **Juergen Strasser**: Conceptualization; Investigation; Methodology; Writing—review and editing. **Johannes Preiner**: Data curation; Supervision; Funding acquisition; Project administration; Writing—review and editing. **Albert JR Heck**: Conceptualization; Resources; Supervision; Funding acquisition; Investigation; Writing—original draft; Project administration; Writing—review and editing.

Source data underlying figure panels in this paper may have individual authorship assigned. Where available, figure panel/source data authorship is listed in the following database record: biostudies:S-SCDT-10_1038-S44318-024-00128-y.

## Disclosure and competing interests statement

The authors declare no competing interests.

# Expanded View Figures

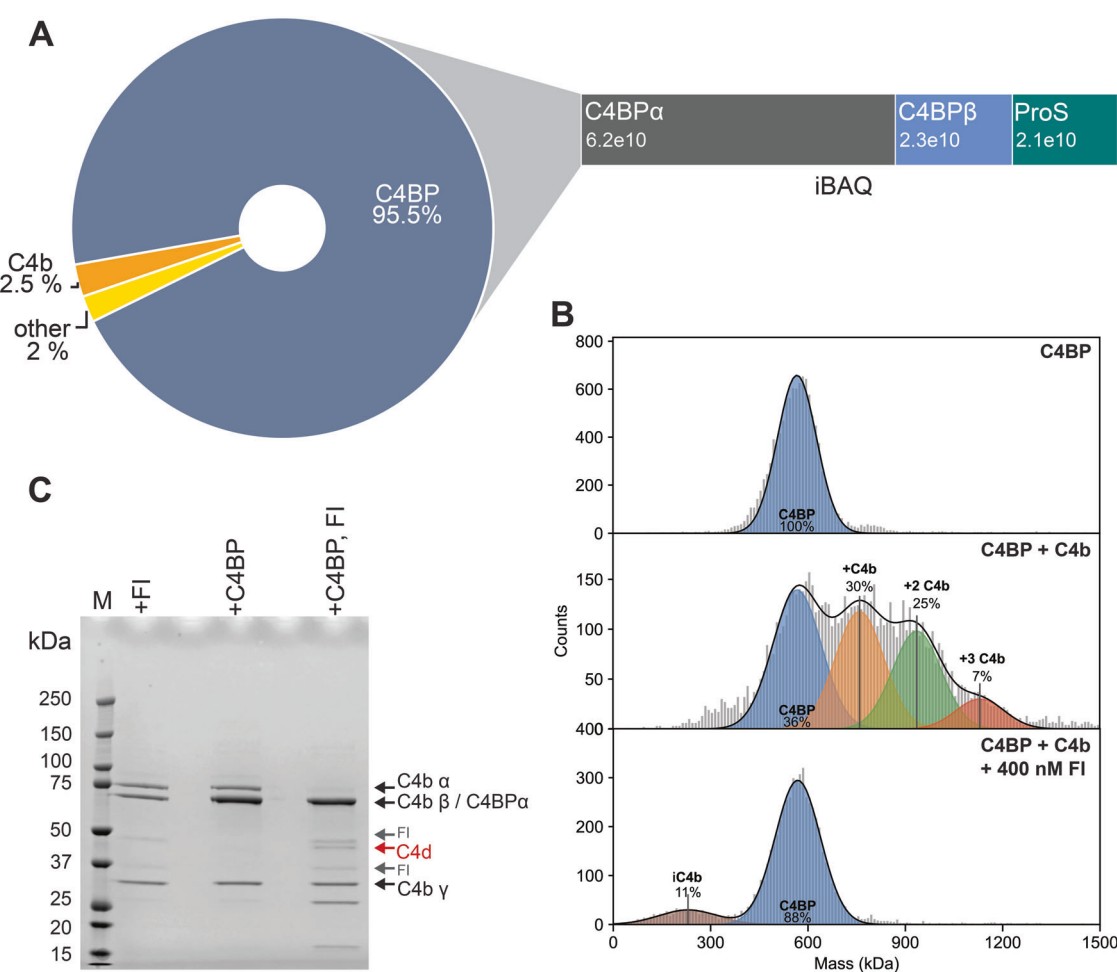

**Figure EV1. C4BP sample composition and activity.**

(A) Composition of the commercial C4BP sample accessed by bottom-up proteomics. The figure shows an abundance of non-contaminant proteins based on their intensity Based Absolute Quantitation (iBAQ) values and suggests that more than 95% of the sample is composed of C4BP HOS formed by C4BPα, C4BPβ interacting with ProS. (B) Mass photometry results highlight C4BP interaction with C4b. The figure shows mass distributions of C4BP (upper panel), C4BP, and C4b in a 1:1 mass ratio (middle panel), and C4BP and C4b in a 1:1 mass ratio with 400 nM FI (bottom panel). The results affirmed C4BPα mediated interaction (Dahlbäck et al, 1983; Blom et al, 2001) resulting in multiple C4b molecules bound to one C4BP. Further, the results show that FI addition leads to C4b degradation (iC4b) and complex disruption. (C) FI-mediated cleavage of C4b in the presence of C4BP. The figure shows the reducing SDS-PAGE of C4b with FI, C4BP, and C4b in a 1:1 mass ratio, and C4BP and C4b in a 1:1 mass ratio with FI, from left to right respectively. The latter shows cleavage of the C4b α-chain by FI in the presence of C4BP. The other two samples display no C4b α-chain showing that C4BP is an essential cofactor in the C4b α-chain degradation. The bands were assigned based on (Blom et al, 2003b). Source data are available online for this figure.

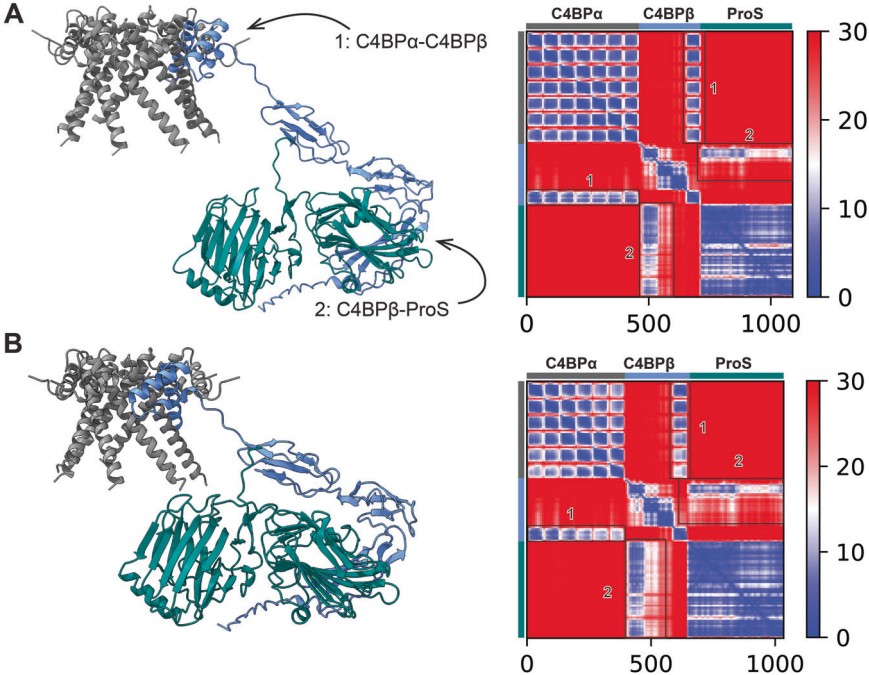

**Figure EV2.  Structural models of C4BP core.**

The figure shows models of the C4BP core generated by AlphaFold-Multimer (v. 2.2.0) and PAE plots for (**A**) α7β1+ProS and (**B**) α6β1+ProS variants. The models show the insertion of C4BPβ into the C4BPα core (area 1 in PAE plot) and the interaction of C4BPβ–ProS (area 2 in PAE plot). The models cover C4BPα (541–597), C4BPβ (1–252), and ProS (284–676). Source data are available online for this figure.

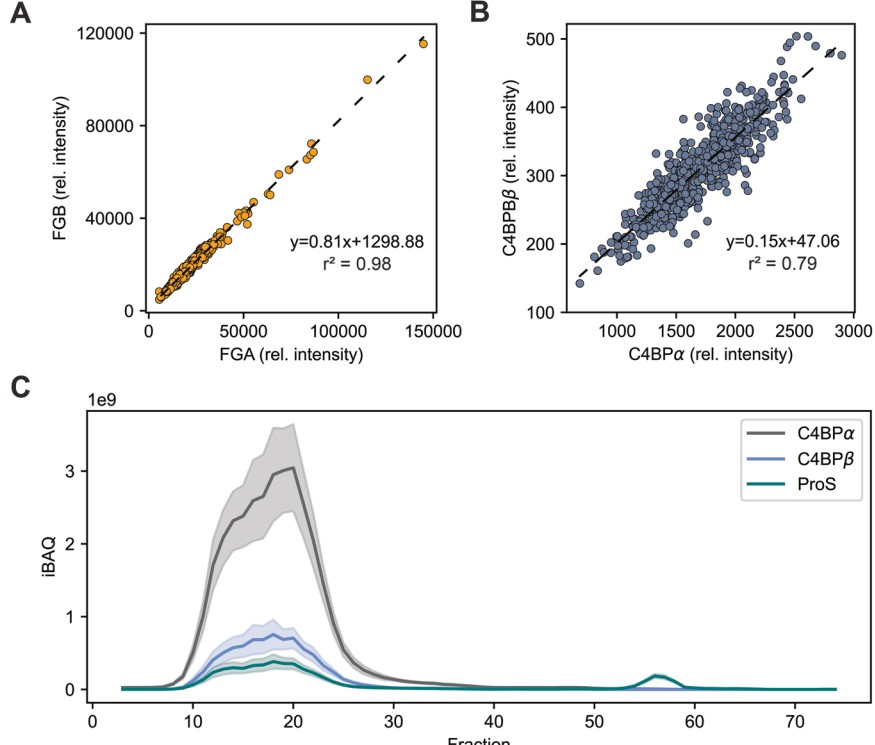

**Figure EV3.  C4BP in human serum.**

(**A**) Correlation of FGA and FGB in a dataset of ~670 human plasma samples (Demichev et al, 2021). The figure shows relative intensities as identified and quantified by DIA-MS proteomics. FGA and FGB relative intensities are shown on the x and y-axis, respectively. The proteins are known to form a stable complex in a 1:1 (Kollman et al, 2009) ratio, which is confirmed by the observed correlation. (**B**) Correlation of C4BPα and C4BPβ in a dataset of ~670 human plasma samples (Demichev et al, 2021). The figure shows relative intensities as identified and quantified by DIA-MS proteomics. Relative intensities of C4BPα and C4BPβ are shown on the x and y-axis, respectively. The figure shows a correlation of C4BPα and C4BPβ relative intensities and indicates more variability in the C4BP assembly compared to the FGA and FGB (**A**). (**C**) Elution profile of C4BPα (gray), C4BPβ (blue), and ProS (teal) in serum SEC. The figure shows pooled healthy donor serum SEC-LC-MS in an experimental triplicate with a standard deviation interval. Fraction numbers are shown on the x-axis and sum iBAQ values are plotted on the y-axis. The results show co-elution of C4BPα, C4BPβ, and ProS in high MW fractions (8–30) affirming their organization in HOS. The low MW fractions (50–60) display elution of unbound ProS supporting the observations in Fig. 7. Source data are available online for this figure.

