## [Peer Review File · The EMBO Journal]

Higher-order structure and proteoforms of co-occurring C4b-binding protein assemblies in human serum

Tereza Kadavá, Johannes Hevler, Sofia Kalaidopoulou Nteak, Victor Yin, Jürgen Strasser, Johannes Preiner, and Albert Heck

Corresponding author: Albert Heck (a.j.r.heck@uu.nl)

Review Timeline:

Submission Date:	15th Dec 23
Editorial Decision:	14th Feb 24
Revision Received:	22nd Mar 24
Editorial Decision:	26th Apr 24
Revision Received:	3rd May 24
Accepted:	8th May 24

Editor: Daniel Klimmeck

Transaction Report:

Dear Albert,

Thank you again for the submission of your manuscript (EMBOJ-2023-116416) to The EMBO Journal and in addition providing us with a preliminary revision plan. Please accept my apologies for getting back to you with unusual protraction due to delayed referee input, as well as detailed discussion in the editorial team. Your study was assessed by four reviewers with expertise in complement biology, structural analyses and proteomics whose comments are enclosed below.

As you will see from their comments, the referees acknowledge the analysis and potential interest and value of your findings. However, they also express major concerns, which need to be addressed thoroughly to make them supportive of publication in the EMBO Journal. In more detail, reviewer #1 raises substantial issues regarding the commercial C4BP used as an experimental source, incomplete annotation of its purification and characterisation of its functionality and purity, which diminishes the relevance of the current findings in his-her view (ref#1, pts.1,2,4,5,7). Referee #2 agrees in that the purification procedures of the commercial C4BP used are not sufficiently specified and requests complementary details (ref#2, pt.1). Reviewer #1 further requests complementary EM characterisation of C4BP (ref#1, pt.10). Further, the reviewers raise a number of points related to the overall structure of the manuscript and presentation of the findings, additional controls required, and overall discussion of related literature, that would need to be conclusively addressed to achieve the level of robustness and clarity needed for The EMBO Journal.

Given the overall interest stated and broader angle of your findings, we are able to invite you to revise your manuscript experimentally to address the referees' comments, along the lines sketched in your outline. I need to stress though that we do require strong support from the referees on a revised version of the study in order to move on to publication of the work. We specifically ask you to address the following issues:

>> complement information on the commercial C4BP sample preparation in the Material & Methods.

>> provide in-depth characterisation of bioactivity and purity of the commercial C4BP sample used.

>> amend the manuscript by complementary analyses based on self-purified C4BP and-or serum-extracted samples for validation and to strengthen support for the claims made on physiological relevance of your structural findings.

Please note that while per se well taken, we concluded the request by referee #1 on additional EM data (ref#1, pt.10) is beyond the scope of the current study thus in our view not required for the revision.

Please feel free to contact me if you have any questions or need further input on the referee comments.

We generally allow three months as standard revision time. As a matter of policy, competing manuscripts published during this period will not negatively impact on our assessment of the conceptual advance presented by your study. However, we request that you contact the editor as soon as possible upon publication of any related work, to discuss how to proceed.

Should you foresee a problem in meeting this three-month deadline, please let me know in advance and I may be able to grant an extension.

When submitting your revised manuscript, please carefully review the instructions below.

Feel free to approach me any time should you have additional questions related to this.

Thank you for the opportunity to consider your work for publication.

I look forward to your revision.

Best regards,

Daniel

Daniel Klimmeck, PhD
Senior Editor
The EMBO Journal

Instruction for the preparation of your revised manuscript:

2) individual production quality figure files as .eps, .tif, .jpg (one file per figure).

3) a .docx formatted letter INCLUDING the reviewers' reports and your detailed point-by-point response to their comments. As part of the EMBO Press transparent editorial process, the point-by-point response is part of the Review Process File (RPF), which will be published alongside your paper.

4) a complete author checklist, which you can download from our author guidelines ([https://wol-prod-cdn.literatumonline.com/pb-assets/embo-site/Author Checklist%20-%20EMBO%20J-1561436015657.xlsx](https://wol-prod-cdn.literatumonline.com/pb-assets/embo-site/Author%20Checklist%20-%20EMBO%20J-1561436015657.xlsx)). Please insert information in the checklist that is also reflected in the manuscript. The completed author checklist will also be part of the RPF.

6) It is mandatory to include a 'Data Availability' section after the Materials and Methods. Before submitting your revision, primary datasets produced in this study need to be deposited in an appropriate public database, and the accession numbers and database listed under 'Data Availability'. Please remember to provide a reviewer password if the datasets are not yet public (see <https://www.embopress.org/page/journal/14602075/authorguide#datadeposition>).

7) Our journal encourages inclusion of *data citations in the reference list* to directly cite datasets that were re-used and obtained from public databases. Data citations in the article text are distinct from normal bibliographical citations and should directly link to the database records from which the data can be accessed. In the main text, data citations are formatted as follows: "Data ref: Smith et al, 2001" or "Data ref: NCBI Sequence Read Archive PRJNA342805, 2017". In the Reference list, data citations must be labeled with "[DATASET]". A data reference must provide the database name, accession number/identifiers and a resolvable link to the landing page from which the data can be accessed at the end of the reference. Further instructions are available at .

8) At EMBO Press we ask authors to provide source data for the main and EV figures. Our source data coordinator will contact you to discuss which figure panels we would need source data for and will also provide you with helpful tips on how to upload and organize the files.

Numerical data can be provided as individual .xls or .csv files (including a tab describing the data). For 'blots' or microscopy, uncropped images should be submitted (using a zip archive or a single pdf per main figure if multiple images need to be supplied for one panel). Additional information on source data and instruction on how to label the files are available at .

9) We replaced Supplementary Information with Expanded View (EV) Figures and Tables that are collapsible/expandable online (see examples in <https://www.embopress.org/doi/10.15252/embo.201695874>). A maximum of 5 EV Figures can be typeset. EV Figures should be cited as 'Figure EV1, Figure EV2' etc. in the text and their respective legends should be included in the main text after the legends of regular figures.

Please remember: Digital image enhancement is acceptable practice, as long as it accurately represents the original data and conforms to community standards. If a figure has been subjected to significant electronic manipulation, this must be noted in the figure legend or in the 'Materials and Methods' section. The editors reserve the right to request original versions of figures and

the original images that were used to assemble the figure.

11) For data quantification: please specify the name of the statistical test used to generate error bars and P values, the number (n) of independent experiments (specify technical or biological replicates) underlying each data point and the test used to calculate p-values in each figure legend. The figure legends should contain a basic description of n, P and the test applied. Graphs must include a description of the bars and the error bars (s.d., s.e.m.).

We realize that it is difficult to revise to a specific deadline. In the interest of protecting the conceptual advance provided by the work, we recommend a revision within 3 months (12th Feb 2024). Please discuss the revision progress ahead of this time with the editor if you require more time to complete the revisions.

Referee #1:

The manuscript submitted by Kadava et al addresses a challenging structural characterization of the C4b binding protein C4BP. Despite considerable progress with respect to structural characterization of many other complement proteins, a detailed overall structure-based understanding of C4BP is lacking due to the presence of multiple isoforms in plasma and the presence of flexible multi-CCP subunits C4BP-alpha and C4BP-beta and the association of the ProS. The aim of increasing our structural comprehension of C4BP is therefore sound and if successful could be of major interest for the field of complement biology.

1) Unfortunately, the authors have not carried out a proper sample preparation prior to conducting sophisticated biophysical and MS based characterization. The sample is apparently commercial C4BP purchased from complement technology (comptech) but this is not addressed properly in the beginning of Results. It is only while reading the Materials and Methods that the reader recognizes where the samples comes from.

2) Authors should provide a full description of the purification protocol employed at comptech such that the reader knows whether the sample has experienced non-physiological buffer conditions prior to MS and biophysical characterization. There is no mention of the ProS in the documents available from CompTech and yet the sample clearly contains ProS.

3) Top of p7, it is stated "The CDMS and MP results clearly showed co-occurring C4BP isoforms in healthy human serum, with the predominant forms being the C4BP(β +) variants $\alpha 7\beta 1$ +ProS and $\alpha 6\beta 1$ +ProS (Fig. 2)."

4) This is not justified by the data presented since we are not aware of the purification history of the C4BP sample if it comes from comptech. Analysis of a true plasma/serum samples from multiple healthy donors with multiple orthogonal methods would be required to justify this statement.

5) In Figure 2 it would be appropriate to describe the purity and oligomerization of the sample using SEC with an accompanying SDS-PAGE analysis of fractions.

6) Authors could also take advantage of SEC-SAXS to complement their MP measurements, this would in addition offer data that could support their later modelling of C4BP

7) A simple experimental functional validation of the C4BP samples (with or without ProS) is also missing, can the sample(s) actually function as cofactor in C4b degradation by FI?

8) Unfortunately, authors do not make a serious attempt to separate the various oligomers present in their commercial sample. Antibodies specific for ProS appears to a very useful tool for subtraction of complexes containing ProS rather than the crude denaturation approach taken by the authors. But even here authors do not document that all ProS is removed from the sample

by a simple SDS-PAGE analysis.

9) Authors to a decent job predicting interfaces between b-subunit and alpha-chain oligomer and b-subunit with ProS using alphafold2 and modelling the glycans of the complexes. Unfortunately, they don't perform an experimental validation using recombinant expression of the wt and mutant full-length or truncated subunits involved of key residues at the intermolecular interface predicted by alphafold.

10) As a more general comment, authors should examine a carefully purified and characterized C4BP by electron microscopy. Recent examples involving also complement proteins argue that even flexible proteins can be imaged at low resolution by EM. It is possible that the core of the complexes can be averaged to give 3D reconstruction of the stable core. With careful reconstitution of truncated subunits it may even be possible to obtain high-resolution structures that reveal the inter-subunit interfaces in details.

Minor issues

P7, top. Define C4BP(+)

On page 9 , authors mention that in their AFM images, additional structures corresponding to EGF-like and Gla domains can be distinguished. This assignment must at best be putative. Authors should consider repeating the experiment in the presence of various Fab fragments with non-overlapping specificities if they really want to map these additional domains in ProS

Legend figure 1. Blom et al 2003a clearly shows that FH is a much more potent cofactor compared to C4BP for C3b degradation. In a physiological setting, FH, CR1 and MCP would be cofactors for C3 degradation, not C4BP. The figure 1 should reflect this in order not to mislead the non-complement reader.

Bottom p10. Rearrangement of ProS domains is mentioned, compared to what?

Referee #2:

General summary and opinion:

In this paper Kadava and colleagues investigate molecular structure of C4b binding protein, its subunit composition, higher order assemblies, and its interaction with protein S. Advanced technologies are used such single particle detection techniques, crosslinking mass spectrometry, glycoproteomics and high-speed atomic force microscopy. The paper builds on the present knowledge of the structure of C4BP and its interaction with protein S but contributes new interesting detailed structural information. The studies focus on the dominating form of C4BP in normal healthy plasma, which is composed of 6-7 alpha-chains and a single beta-chain. Protein S binds to the beta-chain of this form with high affinity. The C4BP-protein S complex has previously been visualized with transmission electron microscopy and its spider-like flexible structure demonstrated.

The present investigation provides high resolution structural details of C4BP alpha chains, its carbohydrate chains, the assembly of the central core with the beta-chain, the interaction between the beta-chain and protein S, demonstrate the high degree of flexibility of the C4BP subunits. It also confirms that during acute phase the plasma levels of C4BP increase, but because the ratio between the alpha- and beta-chains increase the level of beta-chain containing C4BP in plasma remains stable and thus the level of free protein S will be unaffected by the acute phase.

State of the art techniques are used and the experimental work is of high quality. The manuscript is well presented. The conclusions drawn are valid.

Specific major concerns:

I have no major concerns regarding the work that is presented but some suggestions and minor concerns.

1, the C4BP that is used is from a commercial source. The purification technique used is not specified. A standard purification technique often used is based on an initial Ba-citrate absorption which will specifically purify C4BP containing protein S. It is known that around 10% of normal plasma contains C4BP lacking the beta-chain (Dahlbäck B Biochem J 1993 209:847-56). This form increases dramatically during acute phase and may be a dominating form during severe acute phase (Garcia de Frutos et al Blood 1994 84:815-22). The paper gives the impression that the protein S-containing C4BP is the dominating and important form of C4BP. In this context it is of interest that the acute phase protein SAP has been shown to bind to the central core of C4BP - could a possible binding site be identified?

It would be of interest if the authors comment further on the structural difference of the central cores of beta-chain containing C4BP and C4BP lacking the beta-chain.

- 2, In figure 3, several intra-subunit cross-links are shown - is it possible that they are inter-subunit?
- 3, the fourth sentence of the abstract should be rephrased as protein S is not a subunit of C4BP but rather a complexing partner.
- 4, It would be of interest to compare the new structural model of figure 6 to the original em-pictures from 1983 (Dahlbäck et al, PNAS 1983, 80:3461-3465).
- 5, the figure 4 legend ends abruptly.

Referee #3:

This manuscript provides a useful update in the analysis of the complement system regulatory protein C4 - binding protein (C4BP). It illustrates how Protein S (ProS) may interact with C4BP and also provides an analysis of the carbohydrate structures on both proteins.

I have two minor points to raise:

1. I think that the majority of those working on research in the complement system will regard C4BP and ProS as distinct proteins i.e. although ProS, under physiological conditions, can be found to be strongly, non-covalently, linked to C4BP(B+) it is surely not regarded as a subunit of C4BP?

Thus perhaps rephrase the definition of ProS as being a subunit? This point appears several times throughout the text and figure legends - for example

"Human C4BP is a macromolecular glycoprotein composed of three distinct subunits, namely C4BP α , C4BP β , and vitamin K-dependent protein S (ProS), which form an ensemble of coexisting higher-order structures. "

"Three subunits form C4BP higher-order structures in human serum, namely C4BP α in grey, C4BP β in blue and ProS in teal. "

2. Figure 7. This figure adds very little to the structural aspects of the manuscript, and since it involves only two patient samples and two control samples, perhaps it should be omitted until more samples become available?

If Figure 7 is to be retained then the timing of the sampling requires explanation, if comparison is to be made between the different sections of the figure

Not clear exactly what T0 T1T9 denotes in each of the figures. Do they denote the same timeline in each of the sections P1, P2 C1 and C2 within Figure 7

Referee #4:

Heck and co-workers study the structure of complexes involving the complement inhibitor, C4BP, by mass-spectrometry- and non-mass-spectrometry-based methods. First, C4BP complexes isolated from human serum were analyzed by charge detection MS and mass photometry to reveal complexes with different copy numbers of the C4BP alpha subunits (six or seven) with one beta subunit and the interactor protein S (ProS) partially attached. Cross-linking MS using DSS and DMTMM was then used to find interaction regions on the three proteins (C4BPalpha and beta and ProS), which were rationalized on models generated with the help of AlphaFold Multimer predictions. The flexibility of the complex arrangements was confirmed by high-speed AFM. Furthermore, the major glycosylation profiles on the three proteins were characterized by a bottom-up method, and identified glycosylation sites and glycan identities were considered for further structural interpretation.

To apply the insights gained from the structural characterization of C4BP assemblies in a biomedical context, the authors finally compared the complex stoichiometries in serum samples from a longitudinal proteomics study on patients undergoing kidney transplantation and controls. Interestingly, it could be shown that the C4BP alpha/beta ratio appears to change during acute phase inflammation events, while ProS levels remain stable.

In summary, the authors have applied a variety of MS-based methods (CDMS, XLMS, glycoproteomics) to study the organization of native C4BP complexes and have confirmed some of their findings with independent methods such as mass photometry and AFM. The study represents an elegant application of such methods to a target of biomedical/clinical relevance. I do not have major criticisms about this work but suggest that the authors address some minor comments summarized below in a revised version.

Minor comments:

Although cross-links are already visualized on structural models in Figure 3 and are discussed on page 7, the actual model generation is only discussed on page 10. This should be aligned a bit better.

The legend to Figure 4 on page 10 ends abruptly, with at least some part missing.

The method section should be proof-read more carefully, as it contains many poorly worded sentences. Some noteworthy points:

"pythomics" should read "pyteomics" (page 18).

"automated injection time every second" (page 19) - what does that mean?

Why were different FDR cut-offs chosen for the two cross-linking chemistries (page 20), to get more robust statistics?

"trypsin + LysCl" should read "trypsin + LysC" (page 20).

"Bionic" (software) should read "Byonic" (page 21). In the same paragraph, "rare1" and "common2" - are the numbers supposed to be there?

Last lines of page 21: "... signal peptides ... were removed and attached to the core models" - what does that mean, they were removed and added again?

Page 22: I assume 1 microgram and not 1 milligram of sample was analyzed by MS. Same paragraph: The gradient is supposed to go first from 44-99% and then remain constant at 95%? Is this correct?

The reference Kalaidopoulou Nteak et al. should be expanded with more details. Is this a manuscript in preparation, submitted, posted as a preprint etc.?

Figure S1: "XMAS bundle" should read "XMAS package"

Detailed responses to the Referees (referee comments in black, responses in blue)

Referee #1:

The manuscript submitted by Kadava *et al* addresses a challenging structural characterization of the C4b binding protein C4BP. Despite considerable progress with respect to structural characterization of many other complement proteins, a detailed overall structure-based understanding of C4BP is lacking due to the presence of multiple isoforms in plasma and the presence of flexible multi-CCP subunits C4BP-alpha and C4BP-beta and the association of the ProS. The aim of increasing our structural comprehension of C4BP is therefore sound and if successful could be of major interest for the field of complement biology. Unfortunately, the authors have not carried out a proper sample preparation prior to conducting sophisticated biophysical and MS based characterization. The sample is apparently commercial C4BP purchased from complement technology (comptech) but this is not addressed properly in the beginning of Results. It is only while reading the Mat and Methods that the reader recognizes where the samples comes from.

We thank the referee for reviewing our work, and his/her suggestions. We appreciate that the referee acknowledges the importance of characterizing C4BP and its isoforms. We understand that the major concern raised is regarding the commercial C4BP purchased from Complement Technology. We addressed this point thoroughly in the revised manuscript version, making clear we used such a sample at the beginning of Results and Discussion (Fig. EV1). Further, the Methods section was updated to include which sample was used for each analysis. As described below further, we include new data by which we characterized this sample for purity and physiological activity.

Authors should provide a full description of the purification protocol employed at comptech such that the reader knows whether the sample has experienced non-physiological buffer conditions prior to MS and biophysical characterization. There is no mention of the ProS in the documents available from CompTech and yet the sample clearly contains ProS.

The revised manuscript contains a purification protocol in the Methods section (p. 9). Specifically, there were not any non-physiological conditions involved in the purification e.g., no Ba-citrate precipitation. Consequently, ProS was retained in the C4BP HOS due to the native purification procedure, preserving non-covalent interaction between ProS and C4BP β (Dahlbäck and Stenflo, 1981; Dahlbäck *et al*, 1983).

Top of p7, it is stated "The CDMS and MP results clearly showed co-occurring C4BP isoforms in healthy human serum, with the predominant forms being the C4BP(β +) variants α 7 β 1+ProS and α 6 β 1+ProS (Fig. 2)." This is not justified by the data presented since we are not aware of the purification history of the C4BP sample if it comes from comptech. Analysis of a true plasma/serum samples from multiple healthy donors with multiple orthogonal methods would be required to justify this statement. We addressed the sample purification question and complemented the Methods section with details of C4BP purification (p. 9) from serum. Further, we analyzed the sample purity and activity in Fig. 2 and EV1.

Addressing the second part of the comment, we respectfully disagree with the referee. Mass photometry and charge detection mass spectrometry (and HS-AFM) are orthogonal approaches.

Moreover, to expand beyond the data shown in Fig. 2 and 7, and further confirm the relevance of our findings in true serum/plasma samples, we complemented the revised manuscript with Fig. EV3. In Fig. EV3B we reanalyzed a plasma proteomics dataset of more than 650 samples (Demichev *et al*, 2021). The results showed a correlation between the C4BP α and β -chains, affirming their organization in higher-order assemblies with a defined composition. Further, we examined in-house generated serum SEC LC-MS results (Doorduijn *et al*, 2022), which revealed a fitted co-elution of the C4BP α , C4BP β , and ProS chains in high molecular weight fractions, confirming the presence of the C4BP assemblies in serum.

In Figure 2 it would be appropriate to describe the purity and oligomerization of the sample using SEC with an accompanying SDS-PAGE analysis of fractions.

We understand the referee's concerns about the sample purity. The sample oligomerization and purity were analyzed by MP (Fig. 2 and EV1B) and CDMS (Fig. 2). These analyses clearly displayed a sample dominated by C4BP assemblies with no other masses that could be assigned. The sample purity question was further examined by bottom-up proteomics (Fig. EV1A), indicating that more than 95% of the sample is formed by C4BP chains and ProS. Although low in abundance, the most prevalent contamination was C4b (2.5%), a known C4BP interacting partner (Dahlbäck *et al*, 1983). It has been previously shown (de Cordoba *et al*, 1983) that C4BP isoforms cannot be readily separated by SEC and coelute in fractions corresponding to higher molecular weights (with IgM). This is likely related to the hydrodynamic radius of C4BP, which does not appear to be different for the C4BP isoforms. Accordingly, our serum SEC LC-MS dataset analysis displayed C4BP α , C4BP β , and ProS co-elution in high molecular weight fraction (Fig. EV3C). Therefore, MP and CDMS (Fig. 2) are beneficial for resolving the C4BP variants.

Authors could also take advantage of SEC-SAXS to complement their MP measurements, this would in addition offer data that could support their later modelling of C4BP

Such analyses have been previously reported (Perkins *et al*, 1986) and agree with the presented structural models (Fig. 6). The SAXS results suggested that the C4BP "arms" are in solution closer together than the negative stain EM (Dahlbäck *et al*, 1983) and HS-AFM (Fig. 4) suggested. This might also be reflected in our cross-linking MS results, which exhibit self-links connecting C4BP α , and presumably inter-links connecting two C4BP α chains. Therefore, we do not foresee that SEC-SAXS data would contribute substantially to the presented data.

A simple experimental functional validation of the C4BP samples (with or without ProS) is also missing, can the sample(s) actually function as cofactor in C4b degradation by FI?

This is a good point that we could successfully address, we complemented the revised manuscript with Fig. EV1B, C. The results show that the C4BP sample provided by Complement Technology can act as a cofactor in C4b degradation by FI. We showed that C4BP forms complexes with C4b by MP (Fig. EV1B) and facilitates FI-mediated cleavage of C4b α -chain by SDS-PAGE (Fig. EV1C).

Unfortunately, authors do not make a serious attempt to separate the various oligomers present in their commercial sample. Antibodies specific for ProS appears to a very useful tool for subtraction of complexes containing ProS rather than the crude denaturation approach taken by the authors. But even here authors do not document that all ProS is removed from the sample by a simple SDS-PAGE analysis. To remove ProS from the Complement Technology sample, we utilized two approaches. The first was for intact mass analysis, and the second was for HS-AFM. For the CDMS and MP analyses, we used denaturing conditions achieved by FA addition. This approach was used for two main reasons. First, it is fully compatible with both the CDMS and MP analyses. Second, it leads to quantitative disruption of the non-covalent C4BP-ProS interaction, as shown in Fig. 2. Even though there are antibodies against ProS, such interaction would not be quantitative, especially at MP concentrations (<20 nM). Therefore, accurate quantification of C4BP variants would not be possible.

The second approach used for HS-AFM was adapted from a previously published study (de Cordoba *et al*, 1983). The sample was treated with urea, and MWCO filters were used to separate ProS from C4BP and to perform buffer exchange with PBS. We documented the sample using MP (Appendix Fig. S4). The results displayed a sample lacking "unbound" ProS in the low MW range and exhibited the same mass

profile in the high MW range as the FA-treated sample. The full depletion of ProS was further confirmed by HS-AFM (p. 5, High-speed atomic force microscopy, the end of 1st paragraph).

Authors to a decent job predicting interfaces between b-subunit and alpha-chain oligomer and b-subunit with ProS using alphafold2 and modelling the glycans of the complexes. Unfortunately, they don't perform an experimental validation using recombinant expression of the wt and mutant full-length or truncated subunits involved of key residues at the intermolecular interface predicted by alphafold.

This question covers two regions of the C4BP HOS. The first of these is the C4BP β – ProS interface, which we address in the Results and Discussion (p. 7, last 2 paragraphs of Full-length glycosylated models of C4b-binding protein). Mutation analysis of C4BP β suggested the involvement of hydrophobic residues in the V16-F45 region of C4BP β (Webb et al, 2001). The glycosylation influence on the C4BP β – ProS interaction has also been previously analyzed (Lu et al, 1997; Blom et al, 2004). None of the studies showed glycan involvement in the interaction. As shown in Fig. 5 and 6, glycans cover a substantial part of the LG2 domain and LG1-2 interface of ProS, as well as C4BP β CCP domains. These findings contradict the previously proposed interaction interface of C4BP β between the LG1-2 domains of ProS (Blom et al, 2004). In contrast, our C4BP β – ProS model displayed in Fig. 3 and 6 exhibits full compatibility with the mutational analyses.

The second predicted interface of the C4BP assembly is the insertion of C4BP β into the C4BP α core. It has been proposed that Cys 202 and 216 of C4BP β are disulfide-linked to the C4BP α (Kask et al, 2002; Blom et al, 2004; Hofmeyer et al, 2013). Furthermore, it was shown that the N-terminal alpha-helix of C4BP β shares high sequence similarity with C4BP α and is thus predicted to form the oligomerization core. However, no model sufficiently described the interaction. In our opinion, the XL-MS restraints, together with previous observations and sequence homologies, provide sufficient evidence for the interaction. Furthermore, such alpha-helical interfaces are not easily targetable using simple mutational analysis (Abrusán & Marsh, 2016). Therefore, in our opinion such analysis would be difficult to interpret.

As a more general comment, authors should examine a carefully purified and characterized C4BP by electron microscopy. Recent examples involving also complement proteins argue that even flexible proteins can be imaged at low resolution by EM. It is possible that the core of the complexes can be averaged to give 3D reconstruction of the stable core. With careful reconstitution of truncated subunits it may even be possible to obtain high-resolution structures that reveal the inter-subunit interfaces in details

Realizing the benefits of a high-resolution structure, we attempted to perform a cryo-EM analysis of the C4BP sample. However, this was not very successful. This most likely reflects C4BP flexibility and could not be resolved by analyzing C4BP-C4b complexes or even by cross-linking stabilization. Therefore, we employed an integrative structural biology approach to characterize C4BP, which we present in the manuscript.

Minor issues

P7, top. Define C4BP(+)

The sentence in the Introduction part was rephrased to make this clearer: “The α 7 variant is sometimes referred to as C4BP(β -) and, accordingly, ProS-bound isoforms containing C4BP β are noted as C4BP(β +).” (p. 2, 2nd paragraph of Introduction)

On page 9, authors mention that in their AFM images, additional structures corresponding to EGF-like and Gla domains can be distinguished. This assignment must at best be putative. Authors should consider repeating the experiment in the presence of various Fab fragments with non-overlapping specificities if they really want to map these additional domains in ProS.

The assignment was based on the previously published negative stain EM of C4BP (B. Dahlbäck et al, 1983). In their publication, the authors visualized the C4BP assembly structure for the first time and assigned LG1-2 domains of ProS interacting with C4BP β . The HS-AFM data shown in Fig. 4 were obtained at a greater resolution, which enabled us to resolve additional ProS domains that are substantially smaller than LG1-2. This has also been previously suggested (B. Dahlbäck et al, 1983), therefore we assigned the additional ProS domains as EGF-like and GLA domains. To clarify this, we adjusted the sentence (p. 5, 1st paragraph of High-speed atomic force microscopy) accordingly: “Those were, based on earlier reported observations, assigned as LG1 and LG2 domains of non-covalently attached ProS (Dahlbäck et al, 1983). “

Legend figure 1. Blom *et al* 2003a clearly shows that FH is a much more potent cofactor compared to C4BP for C3b degradation. In a physiological setting, FH, CR1 and MCP would be cofactors for C3 degradation, not C4BP. The figure 1 should reflect this in order not to mislead the non-complement reader.

We thank the reviewer for this comment and agree that it might have been misleading. We corrected this in the revised manuscript. The revised Fig. 1 includes FH as the main cofactor for C3b degradation and C4BP in parentheses.

Bottom p10. Rearrangement of ProS domains is mentioned, compared to what?

This sentence referred to the ProS conformation exposing the C4BP β interacting interface. We thank the reviewer for pointing this out, and we agree that the sentence might have been slightly misleading. We omitted it in the revised manuscript version.

Referee #2:

General summary and opinion:

In this paper Kadava and colleagues investigate molecular structure of C4b binding protein, its subunit composition, higher order assemblies, and its interaction with protein S. Advanced technologies are used such single particle detection techniques, crosslinking mass spectrometry, glycoproteomics and high-speed atomic force microscopy. The paper builds on the present knowledge of the structure of C4BP and its interaction with protein S but contributes new interesting detailed structural information. The studies focus on the dominating form of C4BP in normal healthy plasma, which is composed of 6-7 alpha-chains and a single beta-chain. Protein S binds to the beta-chain of this form with high affinity. The C4BP-protein S complex has previously been visualized with transmission electron microscopy and its spider-like flexible structure demonstrated.

The present investigation provides high resolution structural details of C4BP alpha chains, its carbohydrate chains, the assembly of the central core with the beta-chain, the interaction between the beta-chain and protein S, demonstrate the high degree of flexibility of the C4BP subunits. It also confirms that during acute phase the plasma levels of C4BP increase, but because the ratio between the alpha- and beta-chains increase the level of beta-chain containing C4BP in plasma remains stable and thus the level of free protein S will be unaffected by the acute phase.

State of the art techniques are used and the experimental work is of high quality. The manuscript is well presented. The conclusions drawn are valid.

Specific major concerns:

I have no major concerns regarding the work that is presented but some suggestions and minor concerns.

We thank the referee for his/her positive opinion on our work.

1, the C4BP that is used is from a commercial source. The purification technique used is not specified. A standard purification technique often used is based on an initial Ba-citrate absorption which will specifically purify C4BP containing protein S. It is known that around 10% of normal plasma contains C4BP lacking the beta-chain (Dahlbäck B Biochem J 1993 209:847-56). This form increases dramatically during acute phase and may be a dominating form during severe acute phase (Garcia de Frutos *et al* Blood 1994 84:815-22). The paper gives the impression that the protein S-containing C4BP is the dominating and important form of C4BP. In this context it is of interest that the acute phase protein SAP has been shown to bind to the central core of C4BP - could a possible binding site be identified? It would be of interest if the authors comment further on the structural difference of the central cores of beta-chain containing C4BP and C4BP lacking the beta-chain.

We agree that the original manuscript did not adequately discuss the origin and purification of the C4BP sample. Therefore, we included more details in the revised Methods section (p. 9). To answer this question, no Ba-citrate was used in the purification.

To address the second question and further prove the nativity of the C4BP sample (in addition to the results presented in Fig. EV1), we examined C4BP interaction with SAP. We collected MP data for the C4BP-SAP complex, documenting one or two SAP pentamers interacting with one C4BP in the presence of Ca²⁺. We confirmed that the interaction is Ca²⁺ dependent, disrupting it by EDTA addition.

Figure 1 C4BP and its interaction with SAP examined by mass photometry (MP). The figure shows MP of C4BP (upper panel), C4BP-SAP complexes in the presence of Ca²⁺ (middle panel), and disruption of C4BP-SAP complexes by EDTA addition.

To answer the last question, we compared the $\alpha 7\beta 1$ oligomerization core presented in the manuscript with the $\alpha 7$ core (T. Hofmeyer *et al*, 2013). The $\alpha 7\beta 1$ core is nearly 8/7 times larger in diameter (45.4 vs 53 Å) compared to the $\alpha 7$, and the size of $\alpha 6\beta 1$ is comparable to the $\alpha 7$ core. However, in both the $\alpha 7\beta 1$ and $\alpha 6\beta 1$ cores, there are slight differences in the α -chain angles with the horizontal plane, based on its distance from C4BP β . This is not observed for the $\alpha 7$ variant, where all alpha helices are equal.

2, In figure 3, several intra-subunit cross-links are shown - is it possible that they are inter-subunit?

Good point, this may be possible for C4BP α . As all C4BP α chains share the same sequence, it is not feasible to distinguish between C4BP α inter- and intra-links. The only exception is the K595 self-link, which connects two adjacent α -chains. The structural models showed the shortest possible cross-links

with a 2Å difference. This sentence has been added to the Methods section (p. 12, the last paragraph of Cross-linking mass spectrometry data analysis) and the Fig. 3 legend.

3, the fourth sentence of the abstract should be rephrased as protein S is not a subunit of C4BP but rather a complexing partner.

This was adjusted as: "Human C4BP is a macromolecular glycoprotein composed of two distinct subunits, C4BP α and C4BP β . These associate with vitamin K-dependent protein S (ProS) forming an ensemble of coexisting higher-order structures."

4, It would be of interest to compare the new structural model of figure 6 to the original em-pictures from 1983 (Dahlbäck *et al*, PNAS 1983, 80:3461-3465).

As briefly discussed above and in the manuscript (p. 5, 1st paragraph of High-speed atomic force microscopy), our HS-AFM data (Fig. 4) agree with the EM analysis (Dahlbäck *et al*, 1983). However, HS-AFM provided a higher resolution, therefore it allowed for a better comparison with the structural model (Appendix Fig. S2). To highlight the similarity between HS-AFM and EM results, the following sentence was adjusted (p. 6, 3rd paragraph of Full-length glycosylated models of C4b-binding protein): "The resulting full-length glycosylated models of dominant C4BP(β +) variants correspond to the "spider-like" HOS as visualized by HS-AFM (Fig. 4) and previously by negative stain EM (Dahlbäck *et al*, 1983).

5, the figure 4 legend ends abruptly.

This was resolved as (p. 23): "All scale bars correspond to 20 nm."

Referee #3:

This manuscript provides a useful update in the analysis of the complement system regulatory protein C4 - binding protein (C4BP). It illustrates how Protein S (ProS) may interact with C4BP and also provides an analysis of the carbohydrate structures on both proteins.

We thank the referee for this support.

I have two minor points to raise:

1. I think that the majority of those working on research in the complement system will regard C4BP and ProS as distinct proteins i.e. although ProS, under physiological conditions, can be found to be strongly, non-covalently, linked to C4BP(B+) it is surely not regarded as a subunit of C4BP? Thus perhaps rephrase the definition of ProS as being a subunit? This point appears several times throughout the text and figure legends - for example "Human C4BP is a macromolecular glycoprotein composed of three distinct subunits, namely C4BP α , C4BP β , and vitamin K-dependent protein S (ProS), which form an ensemble of coexisting higher-order structures."

"Three subunits form C4BP higher-order structures in human serum, namely C4BP α in grey, C4BP β in blue and ProS in teal. "1

We agree that this was not phrased accurately. To clarify this, we rephrased the following sentences:

- p. 2, Abstract: "Human C4BP is a macromolecular glycoprotein composed of two distinct subunits, C4BP α and C4BP β . These associate with vitamin K-dependent protein S (ProS) forming an ensemble of coexisting higher-order structures."

-p. 12, Methods, Peptide-centric glycoproteomics: "To gain coverage of all C4BP assembly components three different proteolytic digestion workflows were employed, using either trypsin + LysC, trypsin + GluC, or chymotrypsin."

- p. 22, Fig. 1 label: "Two subunits are forming the C4BP higher-order structures in human serum, namely C4BP α in grey and C4BP β in blue. Complexing partner ProS is shown in teal."

-p. 23, Fig. 5 label: "Figure 5. N-glycosylation of C4BP."

2. Figure 7. This figure adds very little to the structural aspects of the manuscript, and since it involves only two patient samples and two control samples, perhaps it should be omitted until more samples become available?

If Figure 7 is to be retained then the timing of the sampling requires explanation, if comparison is to be made between the different sections of the figure

Not clear exactly what T0 T1T9 denotes in each of the figures. Do they denote the same timeline in each of the sections P1, P2 C1 and C2 within Figure 7

To further highlight the relevance of the data shown in Fig. 7, we added Fig. EV3 to the revised manuscript. Throughout the results, we present C4BP from real serum and plasma samples and provide further evidence for the C4BP HOS composition. Specifically, in Fig. EV3B we displayed a correlation of the C4BP α and C4BP β chains in more than 650 human plasma samples, affirming their organization in higher-order assemblies. However, the data showed more variance for the C4BP chains, than fibrinogen FGA and FGB. This observation is somewhat expected and might reflect differences in the C4BP variant composition. In Fig. EV3C we show co-elution of the C4BP chains with ProS in high MW fractions of serum SEC LC-MS dataset. We further observe "unbound" ProS eluting later in lower MW fractions.

To make Fig. 7 more reader-friendly, we added days corresponding to each time point below the time point labels. The sampling and further details regarding the figure are shown in Appendix Fig. S3.

Referee #4:

Heck and co-workers study the structure of complexes involving the complement inhibitor, C4BP, by mass-spectrometry- and non-mass-spectrometry-based methods. First, C4BP complexes isolated from human serum were analyzed by charge detection MS and mass photometry to reveal complexes with different copy numbers of the C4BP alpha subunits (six or seven) with one beta subunit and the interactor protein S (ProS) partially attached. Cross-linking MS using DSS and DMTMM was then used to find interaction regions on the three proteins (C4BPalpha and beta and ProS), which were rationalized on models generated with the help of AlphaFold Multimer predictions. The flexibility of the complex arrangements was confirmed by high-speed AFM. Furthermore, the major glycosylation profiles on the three proteins were characterized by a bottom-up method, and identified glycosylation sites and glycan identities were considered for further structural interpretation.

To apply the insights gained from the structural characterization of C4BP assemblies in a biomedical context, the authors finally compared the complex stoichiometries in serum samples from a longitudinal proteomics study on patients undergoing kidney transplantation and controls. Interestingly, it could be shown that the C4BP alpha/beta ratio appears to change during acute phase inflammation events, while ProS levels remain stable.

In summary, the authors have applied a variety of MS-based methods (CDMS, XLMS, glycoproteomics) to study the organization of native C4BP complexes and have confirmed some of their findings with independent methods such as mass photometry and AFM. The study represents an elegant application of such methods to a target of biomedical/clinical relevance. I do not have major criticisms about this work but suggest that the authors address some minor comments summarized below in a revised version.

We thank the referee for appreciating our work and acknowledge the comments on our manuscript.

Minor comments:

Although cross-links are already visualized on structural models in Figure 3 and are discussed on page 7, the actual model generation is only discussed on page 10. This should be aligned a bit better.

We agree that this section was a bit confusing for the reader. Still, in our opinion the model insights nicely complement the XL-MS data. To clarify this and make the transition smoother, we added following sentence to the Fig. 3 label (p. 22): "The restraints were visualized on C4BP structural models generated as described below in the section Full-length glycosylated models of the C4b-binding protein."

The legend to Figure 4 on page 10 ends abruptly, with at least some part missing.

This has been resolved as (p. 23): "All scale bars correspond to 20 nm."

The method section should be proof-read more carefully, as it contains many poorly worded sentences. Some noteworthy points:

We thank the referee for pointing out this issue. We carefully revised the Methods section, with some changes discussed below.

"pythomics" should read "pyteomics" (page 18).

The typo was corrected (p. 9).

"automated injection time every second" (page 19) - what does that mean?

This sentence, referring to 1 s duty cycle, was not phrased correctly. It was adjusted to (p. 11): "The Orbitrap Exploris 480 collected MS1 scan every second with 60000 resolution, 375-1600 m/z range, standard AGC target, and automatic injection time. Subsequent MS2 scans were performed in data-dependent acquisition mode with a 1.4 m/z isolation window with 14 sec dynamic exclusion after one measurement."

Why were different FDR cut-offs chosen for the two cross-linking chemistries (page 20), to get more robust statistics?

Yes, the DMTMM search is more complex, because the reagent can couple acidic residues E|D with K (compared to the K-K (or N-term) cross-links for DSS).

"trypsin + LysCl" should read "trypsin + LysC" (page 20).

The typo was corrected (p. 12).

"Bionic" (software) should read "Byonic" (page 21). In the same paragraph, "rare1" and "common2" - are the numbers supposed to be there?

The typo was corrected (p. 13). The number refers to the modifications allowed on a single peptide. So, for example, if N-glycans were set as a common2 modification, the tool searched for 0, 1, or 2 glycans on a single peptide.

Last lines of page 21: "... signal peptides ... were removed and attached to the core models" - what does that mean, they were removed and added again?

The sentence was rephrased as (p. 13): “Further, signal peptides of all 3 protein chains were removed, and the truncated protein chains were attached to the core models.”

Page 22: I assume 1 microgram and not 1 milligram of sample was analyzed by MS. Same paragraph: The gradient is supposed to go first from 44-99% and then remain constant at 95%? Is this correct? Yes, there were two typos in the paragraph, and it was correct assumption.

The first typo was corrected, as (p. 14): “Approximately 1 µg of peptides for each sample was analyzed on an Orbitrap Exploris 480 mass spectrometer (Thermo Fisher Scientific) operated in DIA mode coupled to an Ultimate3000 liquid chromatography system (Thermo Fisher Scientific Inc).”

The second sentence was corrected, as (p. 14): “Proteome samples were eluted over a linear gradient of a dual-buffer setup with buffer A (0.1% (v/v) FA) and buffer B (80%(v/v) ACN, 0.1%(v/v) FA) ranging from 9 to 44% B over 65 min, 44–99% B for 3 min, and maintained at 9% B for the final 5 min with a flow rate of 300 nL/min.”

The reference Kalaidopoulou Nteak *et al* should be expanded with more details. Is this a manuscript in preparation, submitted, posted as a preprint etc.?

This was resolved, and the citation was replaced by (Kalaidopoulou Nteak *et al*, 2024). The manuscript is now available as a preprint (<https://www.biorxiv.org/content/10.1101/2024.01.31.578168v1>).

Figure S1: "XMAS bundle" should read "XMAS package"

This is the only point where we respectfully disagree with the referee. ChimeraX packages are officially called bundles, and this applies to XMAS (Lagerwaard et al, 2022).

Additional references:

Abrusán G & Marsh JA (2016) Alpha Helices Are More Robust to Mutations than Beta Strands. *PLOS Computational Biology* 12: e1005242

Dear Dr Albert Heck,

Thank you for submitting your revised manuscript (EMBOJ-2023-116416R) to The EMBO Journal. Your amended study was sent back to the four referees for their scientific re-evaluation, and we have received detailed comments from all of them, which I enclose below. As you will see, the experts state that the work has been substantially improved by the revisions and they are now in favour of publication. Please note that while referee #1's remaining concerns are noted, we have in light of the strong support of the other reviewers decided we can proceed with this study towards acceptance.

Thus, we are pleased to inform you that your manuscript has been accepted in principle for publication in The EMBO Journal.

Please carefully consider referee #1's remaining points and address these by adjusting the discussion and introducing caveats where appropriate.

Also, we now need you to take care of a number of issues related to formatting and data presentation as detailed below, which should be addressed at re-submission.

Please contact me at any time if you have additional questions related to below points.

Thank you for giving us the chance to consider your manuscript for The EMBO Journal. I look forward to your final revision.

Again, please contact me at any time if you need any help or have further questions.

Best regards,

Daniel Klimmeck

>> Adjust the title of the 'Conflict of Interest' statement to 'Disclosure and Competing Interests Statement'.

>> Author Contributions: Please remove the author contributions information from the manuscript text. Note that CRediT has replaced the traditional author contributions section as of now because it offers a systematic machine-readable author contributions format that allows for more effective research assessment. and use the free text boxes beneath each contributing author's name to add specific details on the author's contribution.

More information is available in our guide to authors.

>> Callouts: panels A,B should be called out for Fig 6; there is a callout for Appendix Fig S7 but there is no such figure; callouts are missing for Movies EV1-3.

>> Data Availability Section: please add a URL is provided for the PRIDE dataset; remove the referee token and ensure privacy is released. Please update the Author Checklist accordingly.

>> Source Data: please provide a completed my colleague source data checklist as instructed by my colleague Hannah Sonntag.

>>Please provide the appendix in .pdf file format.

>> Dataset EV legends: The three EV movies should be uploaded separately and each movie should have its legend removed from the manuscript and zipped with the corresponding movie file.

>> Revisit publication status of the bioRxiv - PREPRINT references Graham et al (2019), Kalaidopoulou et al (2024), Evans et al (2022), Lagerwaard et al (2022) and update in case of formal journal publication.

>> Consider additional changes and comments from our production team as indicated below:

- DAS:

1. Please note that the specific URL for PXD047679 dataset is not provided in the data availability statement.
2. Please note that reviewer access code for PXD047679 dataset is provided in the manuscript.

Referee #1:

General comments

The manuscript has been significantly improved and forms a quite valuable addition to our current understanding of C4bp. The manuscript can still be improved significantly by including additional low resolution structural characterization and by properly addressing the initial sample characterization.

Major issues.

Regarding the purification procedure, the revised manuscript states "The procedure included several non-denaturing chromatographic steps without the involvement of denaturing agents and Ba-citrate precipitation." This is not sufficient information to describe the sample purification and certainly not to reproduce it. The only way to reproduce the results of the study is to buy the same preparation from Comptech. Figure EV1 does show C4bp in two SDS-PAGE lanes together with C4b and its degradation products, C4bp purity cannot be judged from this gel. Authors should in figure 2 in the main text (and not in an EV or appendix figure) characterize the commercial sample using at least SEC and preferentially also high performance ion exchange or hydrophobic interaction chromatography if ion exchange for some reason fails. An SDS page analysis of SEC fractions with at least 3 ug in each lane of the peak fractions to allow the reader to truly judge purity of the sample in the absence of C4b should be presented. CDMS and MP cannot replace SEC+SDS-PAGE analysis, which is trivial and standard in the field for sample characterization.

Authors mention a paper from 1983 as reference to argue that the C4bp isoforms are difficult to separate by SEC. Luckily, major improvements in SEC resins took place the past 40 years, authors should attempt again with state of the art columns. Likewise, authors also refer to a SAXS paper by Perkins from 1986 as a reason for not collecting SAXS data. Huge progress has been made in the SAXS field the past 37 years. The Perkins data appears not to extend beyond 1 nm⁻¹, significant data at beyond 2 nm⁻¹ is routine now. Also, if there is separation in SEC, SEC-SAXS is also a very powerful tool. On the modelling side, major progress has been made with respect to integrative modelling that takes into account SAXS data, the modelling presented by Perkins was state of the art in 80ties, but not now. Authors neglect this opportunity to study the solution structure with the most modern approaches. SAXS data offers a unique opportunity for the authors to obtain information about the 3D structure in solution, and it is experimentally feasible in a reasonable time. Since authors are able to provide independent measures of the fraction of the relevant complexes, samples containing a mixture of components like present here can be well analyzed with modern SAXS analysis and modelling tools.

Authors also misunderstood the proposal of using ProS specific antibodies given in the first evaluation. The proposal was to use immobilized antibodies for affinity chromatography to remove C4bp-ProS complexes in a simple manner under native buffer conditions and leave C4bp. This can be done in small scale and should be attempted.

It is admirable that authors attempted cryo-EM but not surprising that they failed in the view of the complexity of C4bp. But a negative stain EM analysis is an easy and fast alternative and may well result in a 3D reconstruction for the central part of C4bp or at least very clear 2D classes. Micrographs presented in Dahlbaek 1983 PMID: 6222381 argue that this is worth the effort. As for SEC and SAXS, huge improvements have occurred in EM data collection and data processing since 1983.

Overall, the manuscript would be significantly stronger if low resolution 3D information could be obtained by SAXS or ns-EM to supplement the beautiful AFM generated pictures. The SAXS data would be able to provide information about the compactness of the particle and its rigidity in solution under native conditions. The models presented in figure 6 would form a

perfect basis for interpretation of SAXS data and ns-EM 2D or 3D classes .

Minor issues. The predicted molecular weight of the relevant species should be given in the text to figure 2.

Lines 115-117 This is C4bp purified from normal human sample, not directly in normal human samples, this should be more clear. One of these isoforms may purify less efficiently by the (secret??) procedure used by comtech.

Line 224-228. The role of glycan on CCP3 is discussed. Authors may want to perform a modelling of the C4b complex based on C3b-FI-FH complex with CCP1-CCP3 fragment to investigate how the glycan could be located relative to C4b and FI.

Referee #2:

The authors have adequately responded to my concerns and revised the manuscript accordingly. I have no further comments or criticism.

Referee #3:

The revised version appears to satisfactorily answer all the major issues raised by the reviewers.

I am happy to recommend acceptance of the manuscript for publication, if it all meets the requirements of the other referees.

Referee #4:

In this revised version, the authors have addressed all my comments appropriately (and I learned something about ChimeraX nomenclature).

The authors addressed the minor editorial issues.

Dear Dr Albert Heck,

Thank you for submitting the revised version of your manuscript. I have now evaluated your amended manuscript and concluded that the remaining minor concerns have been sufficiently addressed.

I am pleased to inform you that your manuscript has been accepted for publication in the EMBO Journal.

Related, I kindly ask for your consent on keeping the referee figure included in this file.

On a different note, I would like to alert you that EMBO Press offers a format for a video-synopsis of work published with us, which essentially is a short, author-generated film explaining the core findings in hand drawings, and, as we believe, can be very useful to increase visibility of the work. Please see the following link for representative examples and their integration into the article web page:

<https://www.embopress.org/doi/full/10.15252/emj.2019103932>

Best regards,

Daniel Klimmeck

Daniel Klimmeck, PhD
Senior Editor
The EMBO Journal
EMBO
Postfach 1022-40
Meyerhofstrasse 1
D-69117 Heidelberg
contact@embojournal.org
Submit at: <http://emboj.msubmit.net>